

# An objective cross-validation framework for mapping rainfall hazard based on rain gauge data

Juliette Blanchet[1], Emmanuel Paquet[2], Pradeebane Vaittinada Ayar[1], and David Penot[2]

[1]Univ. Grenoble Alpes, CNRS, IGE, F-38000 Grenoble, France
[2]EDF – DTG, 21 Avenue de l'Europe, BP 41, 38040 Grenoble Cedex 9, France

**Correspondence:** juliette.blanchet@univ-grenoble-alpes.fr

**Abstract.** We propose an objective framework for estimating rainfall cumulative distribution function within a region when data are only available at rain gauges. Our methodology is based on the evaluation of several goodness-of-fit scores in a cross-validation framework, allowing to assess goodness-of-fit of the full distribution but with a particular focus on its tail. Cross-validation is applied both to select the most appropriate statistical distribution at station locations and to validate the mapping of these distributions. Our methodology is applied to daily rainfall in the Ardèche catchment in South of France, a 2260 km$^2$ catchment with strong disparities in rainfall distribution. Results show preference for a mixture of Gamma distribution over seasons and weather patterns, with parameters interpolated with thin plate spline across this region. However the framework presented in this paper is general and could be likewise applied in any region, with possibly different conclusion depending on the subsequent rainfall processes.

## 1 Introduction

In recent years, Mediterranean storms involving various spatial and temporal scales have hit many locations in southern Europe, causing casualties and damages (Ramos et al., 2005; Ruin et al., 2008; Ceresetti et al., 2012a). Assessing the frequency of occurrence of extreme rainfall in a region is usually done by the computation of return level maps. This requires relating any (large) amount of rainfall at a given location to its return level, i.e. to the frequency such an amount is expected to occur on average at this location. In other words, it requires knowing the cumulative distribution function (CDF) of extreme rainfall at any grid point of the map. However there are other situations when not only the largest rainfalls are of interest, but also smaller and even zero rainfall values. This is for example the case in rainfall simulation frameworks, e.g. when rainfall are input of spatially distributed hydrological models: one needs to be able to simulate any rainfall with the right frequency, and not only the largest ones. Other domains include the evaluation of numerical weather simulations (e.g. Froidurot et al., 2016) or the investigation of the climatology of rainfall events in a region.

A difficulty in producing rainfall return level maps is that knowing the CDF at any grid point ideally requires observing rainfall on a grid scale. However long-enough gridded data with good-enough quality is often lacking. Radar and satellite estimations are usually available for about 10 years at best, and only for selected regions. In addition rainfall estimation in complex topography is particularly tricky, e.g. due to the mountain ranges shielding the radar beam (Germann et al., 2006),



or to the complex relationship between satellite-measured radiances and rainfall reaching the ground (Tian and Peters-Lidard, 2010). On the other hand, rain gauge networks are usually operational for 50 to 100 years in the main part of the world, at least at daily scale, but they only provide point observations. Thus, two main methods are usually adopted for learning the CDF of rainfall at any location when observations are only available at selected locations. The first one resorts to the spatial

interpolation of point data supplied by rain gauges. This allows transforming point observations into gridded ones, and so to estimate gridded CDFs of rainfall. Among the most performing methods for spatial interpolation of daily rainfall are kriging, Inverse Distance Weighting and spline-surface fitting (e.g. Camera et al., 2014; Creutin and Obled, 1982; Goovaerts, 2000; Ly et al., 2011; Rogelis and Werner, 2013). In complex topography, there may be some gain in applying these methods locally, e.g. considering local precipitation altitude gradients (Frei and Schär, 1998; Gottardi et al., 2012; Lloyd, 2010). However none

of the above statistical methods are able to fully account for the statistical properties of rainfall fields. A first difficulty is due to the presence of zeros, which complicates interpolation and can lead to negative interpolated rainfalls - although this could be partially overcome by using analytical transformation of the raw variable. A second difficulty is that rainfall distribution is usually heavy tailed and interpolation methods, by smoothing values, lack quality for representing the most extreme events (Delrieu et al., 2014).

A second way of mapping rainfall hazard is, rather than interpolating the point observations, to map the parameters of CDFs fitted on rain gauge series. In addition to the choice of interpolation models, comes now the choice of the marginal model of rainfall amounts on wet days (referred as nonzero rainfalls). The most commonly used CDFs at daily scale include the exponential, Gamma, lognormal, Pareto, Weibull and Kappa models (Papalexiou et al., 2013). Noting that these distributions tend to underestimate extreme rainfall amounts (Katz et al., 2002), a recent flurry of research developed hybrid models based

on mixtures of distributions for low and heavy amounts (Vrac and Naveau, 2007; Furrer and Katz, 2008; Li et al., 2012). More recently Naveau et al. (2016) proposed a family of distributions that is able to model the full spectrum of rainfall, while avoiding the use of mixtures of distributions. Several studies compared marginal models for rainfall (e.g. Mielke and Johnson, 1974; Swift and Schreuder, 1981; Cho et al., 2004; Husak et al., 2007; Papalexiou et al., 2013), but focusing usually on a couple of CDFs. Other studies compared methods for mapping rainfall hazard, and particularly extreme rainfall, assuming a

given CDF (Beguería and Vicente-Serrano, 2006; Beguería et al., 2009; Szolgay et al., 2009; Blanchet and Lehning, 2010; Ceresetti et al., 2012b). However there is, to the best of our knowledge, no study assessing goodness-of-fit of the *full* procedure of rainfall hazard mapping, i.e. from marginal modeling to the production of hazard maps.

Our study aims at filling this gap by proposing an objective cross-validation framework that is able to validate the full procedure of rainfall hazard mapping starting from point observations. Our framework features three characteristics: i) it

selects both the marginal and mapping models, ii) it validates the full spectrum of rainfall, from small to long-term extrapolated amounts, iii) it applies on a regional scale. The framework is illustrated on the Ardèche catchment in South of France. Despite its relatively small size, this test case is particularly challenging as showing extraordinarily strong disparities in rainfall statistics in very short distance. Section 2 presents the data. Section 3.1 and 3.2 describe the marginal distributions and mapping models considered in this study and present the cross-validation scores of model selection. Section 3.3 detail the procedure of model





selection from marginal modeling to hazard mapping. Section 4 gives extensive results for the Ardèche catchment. Section 5 concludes.

## 2  Data

We illustrate our framework on the Ardèche catchment (2260 km$^2$) located in South of France (see Figure 1). The region
includes part of the south-eastern edge of the Massif Central, where the highest peaks of the region are located (more than 1500 m.a.s.l), and the lower elevated Rhône valley (down to 10 m.a.s.l). The south-eastern slope of the Massif Central is known to experience most of the extreme storms and resulting flash floods (Figure 2 of Nuissier et al., 2008). These so-called "Cévenol" events are produced by quasi-stationary mesoscale convective systems that stabilize over the region during several tens of hours. The positioning and stationarity of these systems are largely influence by the topography of the surrounding
mountain massifs (Nuissier et al., 2008). We use two daily rain gauge networks maintained respectively by Electricité de France and Météo-France. We consider the 15 rain gauges inside the catchment, together with the 27 stations located less than 15km outside. This gives a total of 42 stations with 20 to 64 years of data between January 1, 1948 and December 31, 2013. In both databases, daily values are recorded every day at 6AM UTC, corresponding to rainfall accumulation between 6AM of the previous day and 6AM of the present day.

The Ardèche catchment is chosen for illustration purpose and because, despite its relatively small size, it shows very large disparities in rainfall distribution. Figure 2 shows that a factor 1 to 2.6 is found for the annual totals of daily rainfall and a factor 1 to 3.2 for the annual maxima. for comparison, a factor 1 to 4 is found in annual maxima in the whole of France. For both annual totals and annual maxima, the strongest values in the region concentrate along the Massif Central ridge, while much smaller values are found a few km apart in the Massif Central plateau or in the Piémont. Concentration of daily rainfall
and particularly of extreme daily rainfall along the Massif Central ridge has already been documented in many studies, see e.g. Figure 10 of Blanchet et al. (2016a). We assume in this study temporal stationarity of rainfall. Case of potential nonstationarity due to climate change will be discussed in Section 5.

## 3  Method

### 3.1  Marginal distribution of rainfall

#### 3.1.1  Considered marginal models

Let $R$ be the random variable of daily rainfal amount at a given station. $R$ is zero with probability $p^0$ and, for any $r > 0$, we have the following decomposition:

$$\mathrm{pr}(R \le r) = p^0 + (1 - p^0)G(r), \tag{1}$$

where $G$ is the CDF of nonzero rainfall at the considered station. Choice of $G$ is an issue. One of the difficulty is that we wish to
model adequately both the bulk of the distribution of nonzero rainfall and its tail, i.e. the probability of extreme rainfall to occur.


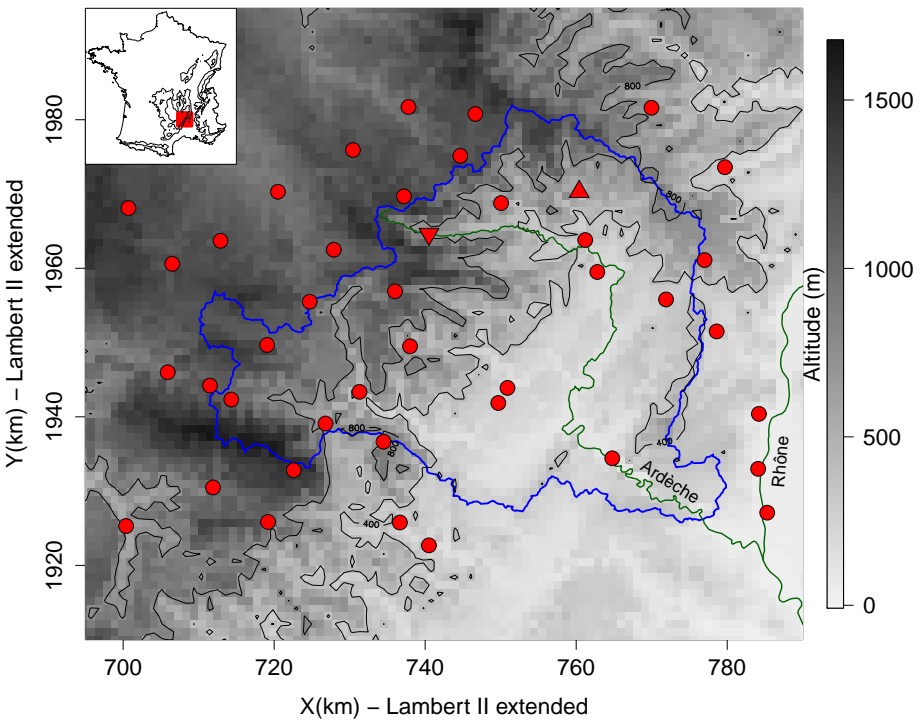

**Figure 1.** Region of analysis. The blue polygon is the Ardèche catchment. The red points show the location of the stations. The upper triangle is the station Antraigues and the lower triangle the station Mayres (both lie at about 500 m.a.s.l.). The background shows the altitude in gray scale (1km raster cells). The top left insert shows a map of France with the studied region in red. The black lines are the 400 and 800 m.a.s.l. isolines.

The most common models for nonzeros rainfall include the Gamma, Weibull and lognormal models (Papalexiou et al., 2013), whose CDF $G(r)$ or densities $g(r) = \partial G(r)/\partial r$, $r > 0$ are given in Table 1. Although less common, another family of models for nonzero rainfall relies on univariate extreme value theory, which tells that probabilities of the form $\mathrm{pr}(R \le r | r > q)$, with $q$ large, can be approximated by either an exponential or a Generalized Pareto tail (Coles, 2001, chapter 4). This led Naveau

5 et al. (2016) to propose the extended exponential and extended Generalized Pareto distributions, whose CDF are given in Table 1. Note that less parsimonious models are given in Naveau et al. (2016) but they are not be considered in the present study. The extended exponential and extended Generalized Pareto distributions of Table 1 insure that the occurrence probability of small (but nonzero) rainfall amounts is driven by $\kappa$ while the upper tail of nonzero rainfall is equivalent to a Generalized Pareto tail. The extended exponential model is also called "Generalized exponential" and it has been used previously for extreme rainfall

10 in Madi and Raqab (2007); Kaźmierczak and Kotowski (2015).





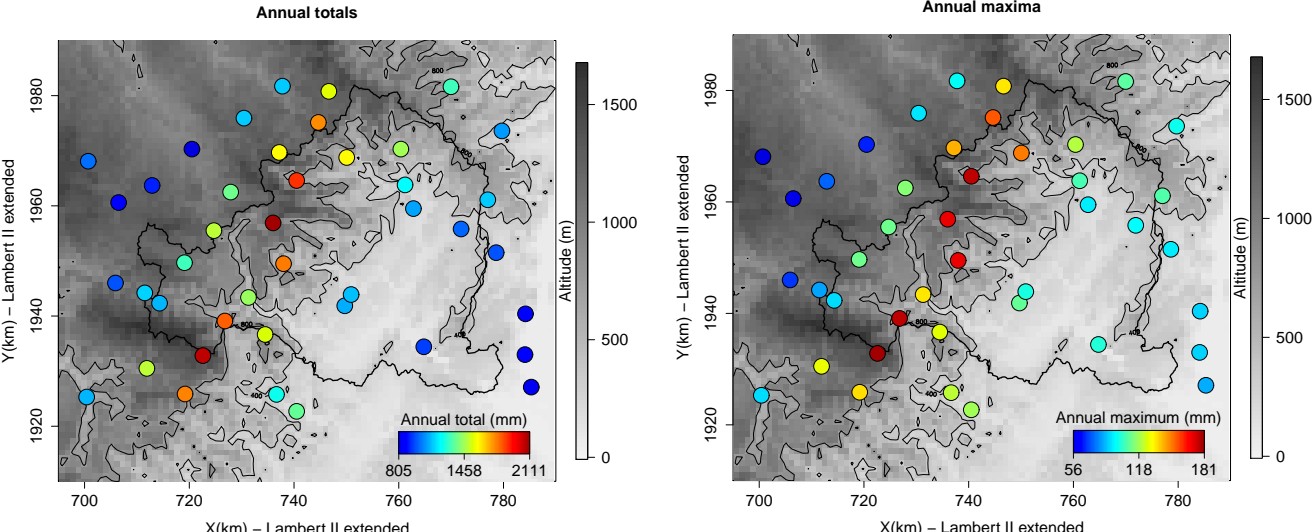

**Figure 2.** Left: Averages of annual totals (mm). Right: Averages of annual maximum daily rainfalls (mm).

| Distribution | CDF $G(r)$ or density $g(r)$, for $r > 0$ | Parameters |
|---|---|---|
| Gamma | $g(r) = (r^{\kappa-1} e^{-r/\lambda})/(\Gamma(\kappa)\lambda^\kappa)$ | $\lambda > 0, \kappa > 0$ |
| Weibull | $G(r) = 1 - e^{-(r/\lambda)^\kappa}$ | $\lambda > 0, \kappa > 0$ |
| lognormal | $g(r) = \exp\{-(\log(r/\lambda))^2/(2\kappa^2)\}/(r\kappa\sqrt{2\pi})$ | $\lambda > 0, \kappa > 0$ |
| extended exponential | $G(r) = (1 - e^{-r/\lambda})^\kappa$ | $\lambda > 0, \kappa > 0$ |
| extended Generalized Pareto | $G(r) = (1 - (1 + \xi r/\lambda)^{-1/\xi})^\kappa$ | $\lambda > 0, \kappa > 0, \xi > 0$ |

**Table 1.** Considered models for the marginal distributions of nonzero rainfall. $\Gamma$ in the Gamma density is the complete Gamma function $\Gamma(\kappa) = \int_0^\infty r^{\kappa-1} e^{-r} dr$

In the models of Table 1, rainfall is implicitly assumed to come from a single distribution. This assumption may be questioned. Indeed, different climatological processes trigger precipitation, leading to the occurrence of rainfall of different natures and intensities (e.g. convective vs. stratiform precipitation). Furthermore, rainfall occurrence and intensities often vary with season, reflecting both variations in temperature and in storm tracks, for example. For this reason, Garavaglia et al. (2010) proposed the use of subsampling based on seasons and weather patterns (WP). Each day of the record period is assigned to a WP. If $S$ seasons and $K$ WP are considered, then days are classified into $S \times K$ subclasses. The law of total probability gives, for all $r > 0$,

$$\text{pr}(R \leq r) = \sum_{s=1}^{S} \sum_{k=1}^{K} \text{pr}(R \leq r | \text{season} = s, \text{WP} = k) \, p_{s,k} \qquad (2)$$





where $p_{s,k}$ is the probability that a given day is in season $s$ and in WP $k$ (thus $\sum_s \sum_k p_{s,k} = 1$). Following (1), $R$ in season $s$ and WP $k$ is zero with probability $p^0_{s,k}$ and, for any $r > 0$, we have the decomposition

$$\text{pr}(R \le r | \text{season} = s, \text{WP} = k) = p^0_{s,k} + (1 - p^0_{s,k})G_{s,k}(r),,$$

where $G_{s,k}$ is the CDF of nonzero rainfall at the considered station for a day in season $s$ and WP $k$. This gives in (2), for all $r > 0$,

$$\text{pr}(R \le r) = p^0 + \sum_{s=1}^{S} \sum_{k=1}^{K} p_{s,k}(1 - p^0_{s,k})G_{s,k}(r), \tag{3}$$

where $p^0 = \sum_{s=1}^{S} \sum_{k=1}^{K} p_{s,k}p^0_{s,k}$ is the probability of any day to be dry. Nonzero precipitation amounts defined by (3) have
CDF:

$$G(r) = \text{pr}(R \le r | R > 0) = \sum_{s=1}^{S} \sum_{k=1}^{K} p'_{s,k}G_{s,k}(r), \tag{4}$$

where $p'_{s,k} = p_{s,k}(1 - p^0_{s,k})/(1 - p^0)$. (4) defines a mixture of $S \times K$ distributions, e.g. a mixture of $S \times K$ Gamma distributions. Similar idea is used in Wilks (1998) for example, but considering a mixture of 2 (exponential) distributions is an unsupervised way, i.e. without relying on a priori subsampling. It shows the advantage of not requiring prior knowledge on the classification
but it is in the same time more difficult to estimate, in particular if the models for different seasons and WP do not differ much.

In this article, we will consider the supervised case (3), with $S = 2$ seasons and $K = 3$ WP, considering the five models of Table 1 for the distribution of precipitation amounts $G_{s,k}$ (see Section 3.3). This implies that estimation of $G_{s,k}$ can be made independently on each others, by considering only the days of the record belonging to season $s$ and WP $k$. A variety of inference methods exists. For rainfall analysis, two options are popular: maximum likelihood (ML) estimation and a method of
moments based on probability weighted moments (PWM). However, as noted in Naveau et al. (2016), ML estimation may fail for rainfall because the discretization due to instrumental precision strongly affects low values, which biaises ML estimation if not accounted for. One way to circumvent this issue is to resort to censored likelihood but choice of the censoring threshold is in itself an issue. Results on our data (not shown) reveal that the threshold has to be no smaller than 5mm. PWM, on the other side, is much more robust against discretization since it is based on summary statistics, rather than on the exact values of
observations (Naveau et al., 2016). For this reason, we estimate in this study the distributions of precipitation amounts $G_{s,k}$ by PWM, while $p^0_{s,k}$ at a given station is estimated empirically as $\hat{p}^0_{s,k} = d^0_{s,k}/d$ where $d$ is the number of observations and $d^0_{s,k}$ is the number of zero values in season $s$ and WP $k$. Combining estimations in (3) gives an estimation of the rainfall CDF at the considered station, and in (4) an estimation of the CDF of nonzero rainfall.

Estimates of return levels are then obtained as follows. The $T$-year return level $r_T$ is the level expected to be exceeded
on average once every $T$ years. It satisfies the relationship $\text{pr}(R \le r_T | R > 0) = 1 - 1/(T\delta)$ where $\delta$ is the mean number of nonzero rainfall per year at the considered station. When subsampling (4) is considered, there is not an explicit formulation and estimation of $r_T$ is obtained numerically by solving $\text{pr}(R \le r_T | R > 0) = 1 - 1/(T\delta)$ in (4).



| Score | Assessment | For which model? |
|-------|------------|------------------|
| NRMSE | Accuracy of the whole distribution | Marginal & mapping models |
| $FF$ | Reliability of the far tail | Marginal & mapping models |
| $N_T$ | Reliability of the close tail | Marginal & mapping models |
| SPAN | Stability at extrapolation | Marginal & mapping models |
| TVD & KLD | Spatial stability | Mapping model |

**Table 2.** Summary of the considered scores for evaluating marginal and mapping models.

### 3.1.2 Evaluation at regional scale in a cross-validation framework

The goal of this evaluation is to assess which marginal model performs better at the regional scale, i.e. for a set of $n$ stations taken as a whole, rather than individually. We follow the split sample evaluation proposed in Garavaglia et al. (2011) and Renard et al. (2013). We divide the data for each station $i$ into two subsamples, $C_i^{(1)}$ and $C_i^{(2)}$, and consider nonzero rainfall for these two subsamples. We fit a given competing model on each of the subsamples, giving two estimated distributions of $G$ in (4): $\hat{G}_i^{(1)}$, estimated on $C_i^{(1)}$, and $\hat{G}_i^{(2)}$, estimated on $C_i^{(2)}$. Our goal is to test the consistency between validation data and predictions of the estimates, both for the core and tail of the distributions, and the stability of the estimates when calibration data changes, focusing particularly on the tail which is usually less stable.

As shown in Table 2, three families of scores are computed, assessing respectively i) accuracy of the estimations along the full range of observations (MEAN(NRMSE)), ii) reliability of the tail of the estimated distribution, checking in particular systematic over or under-estimation of the observations (AREA($FF$) and AREA($N_T$)), and iii) stability of the tail at extrapolation (MEAN(SPAN)). The scores relating the tail of the distribution have been proposed and used in Garavaglia et al. (2011); Renard et al. (2013); Blanchet et al. (2015). In the split sample evaluation framework, four scores can be derived of a given score $Sc$: $Sc^{(12)}$ is the regional score when $G_i^{(2)}$ is validated on the nonzero rainfall subsample $C_i^{(1)}$. $Sc^{(21)}$, $Sc^{(11)}$ and $Sc^{(22)}$ are obtained symmetrically. $Sc^{(11)}$ and $Sc^{(22)}$ are calibration scores, while $Sc^{(12)}$ and $Sc^{(21)}$ are cross-validation scores. For the sake of conciseness, we detail below the case of $Sc^{(12)}$ for the different scores.

The NRMSE (Normalized Root Mean Squared Error) evaluates reliability of the fits in the whole observed range of nonzero rainfall, by comparing observed and predicted return levels of daily rainfall. For a given station $i \in \{1, \ldots, Q\}$,

$$\text{NRMSE}_i^{(12)} = \left\{ \frac{1}{n_i^{(1)}} \sum_{k=1}^{n_i^{(1)}} (r_{i,T_k}^{(1)} - \hat{r}_{i,T_k}^{(2)})^2 \right\}^{1/2} \bigg/ \frac{1}{n_i^{(1)}} \sum_{k=1}^{n_i^{(1)}} r_{i,T_k}^{(1)}, \tag{5}$$

where $n_i^{(1)}$ is the number of nonzero rainfall in $C_i^{(1)}$ for station $i$, $T_k$ ranges the observed return periods of nonzero rainfall in $C_i^{(1)}$, $r_{i,T_k}^{(1)}$ is the observed daily rainfall associated to the return period $T_k$ for the subsample $C_i^{(1)}$ and $\hat{r}_{i,T_k}^{(2)}$ is the $T_k$-year return period derived from the estimated $\hat{G}_i^{(2)}$. Without loss of generality we assume $T_1, \ldots, T_{n_i^{(1)}}$ to be sorted in descending order (so $T_1$ is associated to the maximum over $C_i^{(1)}$). If station $i$ has $\delta_i$ nonzero rainfall per year on average, usual practice is to consider the $k$th largest return period as $T_k = (n_i^{(1)} + 1)/(\delta_i k)$, $k = 1, \ldots, n_i^{(1)}$, and to estimate $r_{i,T_k}^{(1)}$ as the $k$th largest observed





rainfall over $C_i^{(1)}$. Estimate $\hat{r}_{i,T_k}^{(2)}$ is obtained numerically from $\hat{G}_i^{(2)}$ as described in Section 3.1. The normalization by the mean rainfall of $C_i^{(1)}$ in (5) allows comparison of NRMSE over stations with different pluviometry. The smaller $\mathrm{NRMSE}_i^{(12)}$, the better $\hat{G}_i^{(2)}$ fits the rainfalls over $C_i^{(1)}$. For the set of $Q$ stations, we obtain a vector of $\mathrm{NRMSE}^{(12)}$ of length $Q$ which should remain reasonably close to zero. A regional score is obtained by computing the mean of the $Q$ values:

$$5 \quad \mathrm{MEAN}(\mathrm{NRMSE}^{(12)}) = \frac{1}{Q} \sum_{i=1}^{Q} \mathrm{NRMSE}_i^{(12)}. \qquad (6)$$

For competing models, the closer the mean is to $0$, the better the goodness-of-fit.

NRMSE assesses goodness-of fit of the whole distribution in the observed range. Now let have a closer look at the tail of the distribution, and in particular at the maximum over $C_i^{(1)}$, i.e. at $r_{i,T_1}^{(1)}$ in (5), that for shortness we denote $m_i^{(1)}$. If $G_i$ is the true distribution of nonzero rainfall, then the corresponding random variable $M_i^{(1)}$ has distribution $G_i$ to the power $n_i^{(1)}$, whose variance is large. Thus computing error based on the single realization $m_i^{(1)}$ would be very uncertain. For this reason, Renard et al. (2013) proposed to make evaluation by pulling together the maxima of the $Q$ stations, after transformation to make them on the same scale. It is based on the idea that if $X$ has CDF $F$, then $F(X)$ follows the uniform distribution on $(0,1)$. Taking $X = M_i^{(1)}$ and $F = (G_i)^{n_i^{(1)}}$ implies that, if $\hat{G}_i^{(2)}$ is a perfect estimate of $G_i$ then

$$ff_i^{(12)} = \{\hat{G}_i^{(2)}(m_i^{(1)})\}^{n_i^{(1)}}$$

should be a realization of the uniform distribution. For the set of $Q$ stations, this gives a uniform sample $ff^{(12)}$ of size $Q$. Hypothesis testing for assessing the validity of the uniform assumption is challenging because the $ff_i^{(12)}$ are not independent from site to site, due to the spatial dependence between data. Thus Blanchet et al. (2015) proposed to base comparison on the divergence of the density of the $ff^{(12)}$ to the uniform density. A reasonable estimate of the latter is obtained by computing the empirical histogram of the $ff^{(12)}$ with 10 equal bins between 0 and 1. As illustrated in Figure 3, if $\hat{G}_i^{(2)}$ are good estimates of $G_i$, $i = 1, \ldots, Q$, the histogram of $ff^{(12)}$ should be reasonably uniform on $(0,1)$. If the histogram is left-skewed, then $\hat{G}_i^{(2)}(m_i^{(1)})$ tends to overestimate the true $G_i^{(1)}(m_i^{(1)})$, or in other words the return period of the maximum over $C_i^{(1)}$ tends to be underestimated (case of over-estimated risk). If the histogram is right-skewed, the return period of the maximum over $C_i^{(1)}$ tends to be over-estimated (case of under-estimated risk). Although any scenario of misfitting could theoretically be possible, in practice the histograms of $ff^{(12)}$ show mainly the three above alternatives: either a good fit (flat histogram), or a tendency towards a systematic under- or over-estimation (left- or right-skewed histograms). By focusing of maximum values, the histogram of $ff^{(12)}$ can be seen as a way of assessing systematic bias in the very tail of the distribution. For a more quantitative assessment, we compute the area between the density of the $ff^{(12)}$ and the uniform density as follows:

$$20 \quad \mathrm{AREA}(FF^{(12)}) = \frac{1}{18} \sum_{c=1}^{10} \left| 10 \frac{\#\{ff_i^{(12)} \in \mathrm{bin}(c), i = 1, \ldots, n\}}{n} - 1 \right|, \qquad (7)$$

where $\#$ is the number of elements of the set. The term inside the absolute value in (7) is the difference between densities in the $c$th bin. The division by $18$ forces the score to lie in the range $(0,1)$ with lower values indicating better fits (the worst case being all values lying in the same bin). Figure 3 shows that, when $Q = 42$ stations are considered, a value of $\mathrm{AREA}(FF^{(12)})$





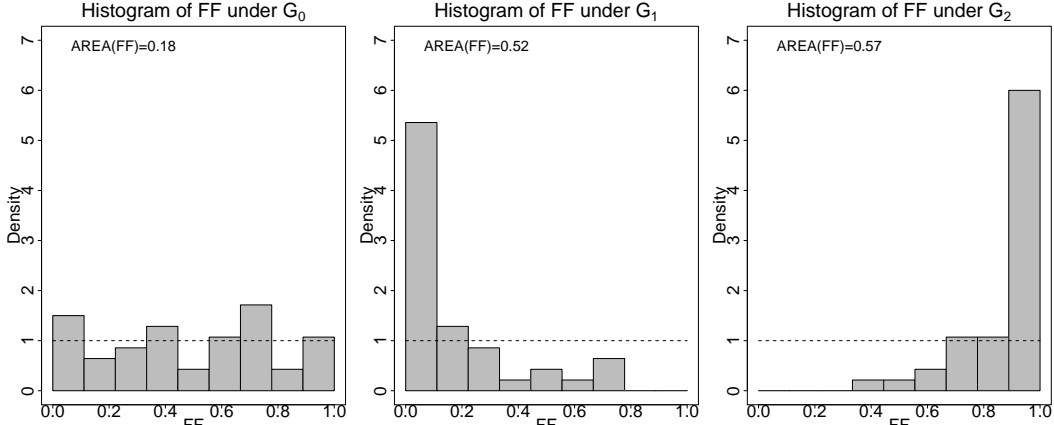

**Figure 3.** Illustration of the $FF$ score when the true CDF $G_0$ is extended exponential with $\lambda = 20$ and $\kappa = 0.3$. The CDF $G_1$ underestimates $G_0$ ($\lambda = 25$) while $G_2$ overestimates $G_0$ ($\lambda = 15$). Left: Histogram of $\{G_0(m)\}^n$ where $m$ are 42 realizations of $G_0^n$ and n=4000. Middle: Histogram of $\{G_1(m)\}^n$. Right: Histogram of $\{G_2(m)\}^n$. The horizontal dashed lines show the uniform density on $(0, 1)$.

around $0.2$ corresponds to no systematic bias in the very tail of the distribution at regional scale, whereas a value around $0.5$ corresponds to a strong over- or under-estimation. In the latter case, only looking at the histogram can inform on whether over- or under-estimation applies.

The $N_T$ criterion is an alternative to $FF$ assessing reliability of the fit of the tail but focusing on prescribed (large) quantiles rather than on the overall maximum. It applies the same principle as $FF$, involving a transformation of $X$ to $F(X)$, but considering $X$ as $K_{i,T}^{(1)}$, the random variable of the number of exceedances over $\mathrm{C}_i^{(1)}$ of the $T$-year return level, i.e. $K_{i,T}^{(1)} = \#\{R_{i,j} \in \mathrm{C}_i^{(1)}; G_i(R_j) > 1 - 1/(T\delta_i)\}$, in which case $F$ is the Binomial distribution $B_i^{(1)}$ with parameters $(n_i^{(1)}, 1/(T\delta_i))$. Thus if $\hat{G}_i^{(2)}$ is a perfect estimate of $G_i$ then $n_{i,T}^{(12)} = B_i^{(1)}(k_{i,T}^{(12)})$, where

$$k_{i,T}^{(12)} = \#\{r_{i,j} \in \mathrm{C}_i^{(1)}; \hat{G}_i^{(2)}(r_{i,j}) > 1 - 1/(\delta_i T)\},$$

should be a realization of the discrete uniform distribution. Randomisation to transform $n_{i,T}^{(12)}$ to a continuous uniform variate on $(0, 1)$ is proposed in Renard et al. (2013) and extensively described in Blanchet et al. (2015). For $i$ ranging over the set of $Q$ stations, we thus obtain a sample of $Q$ uniform variates. Scores are calculated as for $FF$ by comparing the empirical densities of $N_T^{(12)}$ to the theoretical uniform density, giving the scores $\mathrm{AREA}(N_T^{(12)})$. Taking $T$ as e.g. half to one-quarter the length of the observations allows to assess reliability of the close tail of the distribution. As such, it is a good complement to $FF$ that
focuses on the far tail (i.e. on the maximum).

Last but not least, the SPAN criterion evaluates the stability of the return level estimation, when using data for each of the two subsamples. More precisely, for a given return period $T$ and station $i$,

$$\mathrm{SPAN}_{i,T} = \frac{|\hat{r}_{i,T}^{(1)} - \hat{r}_{i,T}^{(2)}|}{1/2\{\hat{r}_{i,T}^{(1)} + \hat{r}_{i,T}^{(2)}\}}, \tag{8}$$





where $\hat{r}_{i,T}^{(1)}$, e.g., is the $T$-year return level for the distribution $G$ estimated on subsample $C_i^{(1)}$ of station $i$, i.e. such that $\hat{G}_i^{(1)}\{\hat{r}_{i,T}^{(1)}\} = 1 - 1/(T\delta_i)$. $\text{SPAN}_{i,T}$ is the relative absolute difference in $T$-year return levels estimated on the two subsamples. It ranges between 0 and 2; the closer to 0, the more stable the estimations for station $i$. For the set of $Q$ stations, we obtain a vector of $\text{SPAN}_T$ of length $Q$ with a distribution which should remain reasonably close to zero. A rough summary of this

information is obtained by computing the mean of the $Q$ values of $\text{SPAN}_{i,T}$, $i = 1, \ldots, Q$:

$$\text{MEAN}(\text{SPAN}_T) = \frac{1}{Q} \sum_{i=1}^{Q} \text{SPAN}_{i,T}. \tag{9}$$

For competing models, the closer the mean is to 0, the more stable is the model. When $T$ is larger than the observed range of return periods, $\text{MEAN}(\text{SPAN}_T)$ evaluates the stability of the return levels in extrapolation. Note that it is by definition 0 in calibration and thus it is only useful in cross-validation.

For the sake of concision, in the rest of this article the scores $\text{MEAN}(\text{NRMSE})$, $\text{AREA}(FF)$, $\text{AREA}(N_T)$ and $\text{MEAN}(\text{SPAN}_T)$ will be referred to as the NRMSE, $FF$, $N_T$ and $\text{SPAN}_T$ scores.

## 3.2 Mapping of the margins

### 3.2.1 Considered mapping models

Let $R_i$ be the random variable of daily rainfall at station $i$, $i = 1, \ldots, Q$. Applying Section 3.1 at station $i$ gives an estimate $\hat{G}_i(r)$

of the CDF $G_i(r) = \text{pr}(R_i \leq r | R_i > 0)$. Our goal is to derive an estimate of the CDF of nonzero daily rainfall at any location $l$ of the region, $\text{pr}(R(l) \leq r | R(l) > 0)$, based on the $Q$ estimated CDFs $\hat{G}_i(r)$. Location $l$ refers here to the three coordinates of ground projection coordinates and altitude, that we write $l = (x, y, z)$. Let $\hat{\theta}_i$ be the set of estimated parameters for station $i$ and $\hat{\theta}_{i,j}$ its $j$th element. $\hat{\theta}_i$ is composed of the $S \times K$ probability of zero rainfall $p_{s,k}^0$ and the $2 \times S \times K$ or $3 \times S \times K$ parameters of the distributions $G_{s,k}$, depending on the marginal distribution (see Table 1). We assume the $\theta_{i,j}$ ordered so that the first

$S \times K$ elements are the $p_{s,k}^0$. We aim at estimating the surface response $\theta_j(l)$ at any $l$ of the region, knowing $\theta_j(l_i) = \hat{\theta}_{i,j}$. In this study we consider three of the most popular method: kriging interpolation, linear regression methods and thin plate spline regressions. However the parameters $\theta_j$s are constrained whereas these models apply the unbounded variables: the probabilities $p_{s,k}^0$ lie in $(0, 1)$, while the parameters of Table 1 are all positive. Therefore we apply the mapping models to transformations of $\theta_j$, i.e. to $\psi_j = tr(\theta_j)$ where $tr$ maps the range of values of $\theta_j$ to $(-\infty, +\infty)$. In this study we consider $\psi_j(l) = \Phi\{\theta_j(l)\}$

if $j \leq S \times K$ (i.e. if $\theta_j$ is any $p_{s,k}^0$), where $\Phi$ is the standard Gaussian CDF, and to $\psi_j(l) = \log\{\theta_j(l)\}$ otherwise. Other transformations would be possible, in particular $p_{s,k}^0$ may be transformed with the logit function, but will not be considered here for the sake of concision. Thus we aim at estimating $\tilde{\psi}_j(l)$ given values $\psi_j(l_i) = \hat{\psi}_{i,j}$ at station locations, with obvious notations. If $l \leq S \times K$, estimates of $\theta_j(l)$ are then obtained as $\tilde{\theta}_j(l) = \Phi^{-1}(\tilde{\psi}_j(l))$. Otherwise surface response estimates are obtained as $\tilde{\theta}_j(l) = \exp(\tilde{\psi}_j(l))$. For the sake of clarity, we omit below the index $j$, considering a surface $\psi(l)$ to be estimated

for all $l$ in the region, given values $\psi(l_i) = \hat{\psi}_i$.

The considered mapping models are listed in Table 3. Three families of method are considered: kriging, linear regression and thin plate spline. Additionally to how they map values, there is a fundamental difference between these models: kriging is





| Name | Model | Coordinates | exact? |
|------|-------|-------------|--------|
| krig | Kriging without external drift | $x, y$ | yes |
| krigz | Kriging with external drift | $x, y, z$ | yes |
| krigZ | Kriging with external drift | $x, y, Z$ | yes |
| steplmz | Stepwise linear regression | $x, y, z$ | no |
| steplmZ | Stepwise linear regression | $x, y, Z$ | no |
| tps2 | Bivariate thin plate spline | $x, y$ | no |
| tps2z | Bivariate thin plate spline with drift | $x, y, z$ | no |
| tps2Z | Bivariate thin plate spline with drift | $x, y, Z$ | no |
| tps3z | Trivariate thin plate spline | $x, y, z$ | no |
| tps3Z | Trivariate thin plate spline | $x, y, Z$ | no |

**Table 3.** Mapping models considered in this study, with involved coordinates. Kriging method provides exact interpolation, unlike the linear regression and thin plate spline.

an exact interpolation, i.e. $\tilde{\psi}(l_i) = \hat{\psi}_i$ at any station location $l_i$ used to estimate the model. On the contrary the linear regression models and thin plate splines provide inexact interpolations: in the great majority of the time, $\tilde{\psi}(l_i) \neq \hat{\psi}_i$ (the goal being obviously to minimize the overall error).

For the kriging interpolation, cases with and without external drift are tested (chapter 3.6 of Diggle and Ribeiro, 2007). The external drift, if any, is modeled as a linear function of altitude (i.e. of the form $a_0 + a_1\zeta$), considering $\zeta$ as either the altitude of the station ($z$), or, following Hutchinson (1998), as a smoothed altitude ($Z$) derived by smoothing a 1km Digital Elavation Model (DEM) with 5km moving windows (i.e. taking $Z$ as the average altitude of 25 DEM grid points). The results that will be presented below correspond to the case of an exponential covariance function of the form $\rho(h) = e^{-h/\beta}$, with $\beta > 0$. We also considered the case of a powered exponential covariance function $\rho(h) = e^{-(h/\beta)^\nu}$ with $0 < \nu \leq 2$, but this resulted in a slight loss of stability due to the additional degree of freedom, without improving the accuracy at regional scale. For the sake of concision, these results are not presented here. Combining alternatives for the drift part gives a total of three kriging interpolation models with 2 to 3 unknown parameters each, for each $\psi$. Estimation of the kriging models are made by maximizing the likelihood associated to the $\hat{\psi}_i$, assuming that $\psi(l)$ is a Gaussian process (see chapter 5.4 of Diggle and Ribeiro, 2007). Alternatives are to estimate drifts and variograms by least squares in different steps, with the risk of biaising estimates (chapter 5.1 to 5.3 of Diggle and Ribeiro, 2007). Both estimation methods are available in the R package 'geoR' (e.g. functions 'likfit' and 'variofit'). In the case without drift, prediction at any site $l$ of the region is obtained as

$$\tilde{\psi}(l) = \sum_{i=1}^{Q} w_i(h_i) \hat{\psi}_i, \tag{10}$$

where the weights $w_i(h_i)$ are derived from the kriging equations and satisfy $\sum_{i=1}^{Q} w_i(h_i) = 1$. The weights depend on the estimated covariance function and on the distance $h_i$ between $l$ and station location $l_i$ in the $(x, y)$ space (i.e. $h_i^2 = (x - x_i)^2 +$

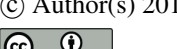


$(y - y_i)^2)$. In the case with external drift, prediction at any location $l$ of the region is then obtained as

$$\tilde{\psi}(l) = a_1\zeta + \sum_{i=1}^{Q} w_i(h_i)(\hat{\psi}_i - a_1\zeta_i),\tag{11}$$

where $\zeta$ is the altitude at location $l$ (i.e. either $z$ or $Z$). Predictions (10) and (11) are exact: $\tilde{\psi}(l_i) = \hat{\psi}_i$, and consequently $\tilde{\theta}(l_i) = \hat{\theta}_i$.

For the linear regression models, we start from regressions of the form $\psi(l) = a_0 + a_1x + a_2y + a_3\zeta + a_4x^2 + a_5y^2 + a_6xy + a_7x\zeta + a_8y\zeta + \epsilon(l)$, where $\epsilon(l) \sim N(0, \sigma^2)$ and $\zeta$ is, as before, either the altitude of the station ($z$) or the smoothed altitude ($Z$). We consider Akaike Information Criteria (AIC), defined as AIC$= 2\eta - 2\log L$, where $\eta$ is the number of parameters (10 at the start) and $L$ is the maximum likelihood value of the regression model. The lower AIC, the better the model. Then we repeatedly drop the variable that increases most the AIC -if any-, and add the variable that decreases most the AIC -if any. This

stepwise method is implemented in the 'step' function of the R package 'stats'. At algorithm stop, the model may contain 1 to 10 parameters, for each $\psi$. Predictions $\tilde{\theta}(l)$ are then obtained as the back-transformation the estimated regressions.

Last but not least, bivariate and trivariate thin plate splines are considered for $\psi(l)$ (Boer et al., 2001; Hutchinson, 1998). These methods are implemented in the function 'Tps' of the R package 'fields'. In the bivariate case, $\psi(l)$ is modeled as $\psi(l) = u(x,y) + \epsilon(x,y)$ where $u$ is an unknown smooth function and $\epsilon(x,y)$ are uncorrelated errors with zero means and equal

variances. The function $u$ is estimated by minimizing

$$\sum_{i=1}^{Q}(\hat{\psi}_i - u(x_i, y_i))^2 + \lambda \int_{-\infty}^{+\infty}\int_{-\infty}^{+\infty}\left\{(\frac{\partial^2 u(x,y)}{\partial x^2})^2 + 2(\frac{\partial^2 u(x,y)}{\partial x\partial y})^2 + (\frac{\partial^2 u(x,y)}{\partial y^2})^2\right\}dx\,dy,\tag{12}$$

where $\lambda$ is the so-called smoothing parameter, which controls the trade-off between smoothness of the estimated function and its fidelity to the observations. It can be estimated by generalized cross-validation. Then predictions are obtained as

$$\tilde{\psi}(l) = a_0 + a_1x + a_2y + \sum_{i=1}^{Q} b_i h_i^2 \log(h_i),\tag{13}$$

where $h_i$ is the Euclidean distance in the $(x,y)$ space between $l$ and station location $l_i$. The partial trivariate case assumes that $\psi(l) - a_3\zeta$ is a bivariate thin plate spline, where $a_3$ is fixed and $\zeta$ is either $z$ or $Z$. To make the connection with kriging, $\psi(l)$ can thus also be seen as a bivariate thin plate spline with (fixed) drift in $\zeta$. The coefficient $a_3$ is estimated in a preliminary step by regressing $\hat{\psi}_i$ against $\zeta_i$. Estimation of the bivariate thin plate spline for $\psi(l) - a_3\zeta$ is made as described above given the values of $\hat{\psi}_i - a_3\zeta_i$. Predictions are obtained as

$$\tilde{\psi}(l) = a_0 + a_1x + a_2y + a_3\zeta + \sum_{i=1}^{Q} b_i h_i^2 \log(h_i),\tag{14}$$

Finally in the trivariate case, we have $\psi(l) = u(x,y,\zeta) + \epsilon(x,y,\zeta)$. The minimization problem is similar to (12) with a penalization enlarged by several terms (Wahba and Wendelberger, 1980). Predictions are then obtained as

$$\tilde{\psi}(l) = a_0 + a_1x + a_2y + a_3\zeta + \sum_{i=1}^{Q} b_i h_i',\tag{15}$$

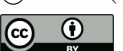



where $h'_i$ is the Euclidean distance in the $(x, y, \zeta)$ space between $l$ and station location $l_i$, scaling the altitude by a factor of 10 following Boer et al. (2001); Hutchinson (1998) (i.e. $h'^2_i = (x - x_i)^2 + (y - y_i)^2 + 100(\zeta - \zeta_i)^2$). Coefficients $a_i$ and $b_i$ in (13) to (15) are estimated by solving a linear system of order $Q$ involving the smoothing parameter $\lambda$. Note that the trivariate case (15) differs from the bivariate case with drift (14) in two ways. First, (15) considers distance in the $(x, y, \zeta)$ space whereas (14)

considers distance in the $(x, y)$ space. Second, in (15) the weights associated to the stations are linear functions of the distance, unlike in (14) (see the term $h^2_i \log(h_i)$ vs. $h'_i$).

### 3.2.2   Evaluation at regional scale in a cross-validation framework

Evaluation is performed in two ways. The first one is a leave-one-out cross-validation scheme aiming to test at regional scale how the interpolated distributions are able to fit the data of the stations when these data are left out for estimating the mapping

model. The second step assesses spatial stability by comparing the interpolated distributions obtained at a given station whether the data of this station are used or not in the mapping estimation. In other words, it is a comparison between leave-one-out and leave-zero-out mappings. So the two evaluations differ in that the first one compares an interpolated distribution to data, while the second step compares two interpolated distributions.

First, let consider a given parameter estimate $\hat{\theta}^{(1)}_{i,j}$ obtained at station $i$ based on the subsample $C^{(1)}_i$ (for a given marginal

model). We apply a leave-one-out cross-validation scheme: for each station $i_0$ alternatively, we use the set of $\hat{\theta}^{(1)}_{i,j}$, $i \neq i_0$ to estimate the surface response $\tilde{\theta}^{(1)}_j(l)$. We store the value of this estimate at station location $i_0$, i.e. $\tilde{\theta}^{(1)}_j(l_{i_0}) = \tilde{\theta}^{(1)}_{i_0,j}$ Repeating this for every parameter $\theta_j$ gives estimation of the full set of parameters at station $i_0$, i.e. estimation $\tilde{G}^{(1)}_{i_0}$ of $G_{i_0}$. Iterating over the stations, we obtain new estimates $\tilde{G}^{(1)}_1, \ldots, \tilde{G}^{(1)}_Q$ of $G_1, \ldots, G_Q$. Applying similarly for the subsample $C^{(2)}_i$ gives estimates $\tilde{G}^{(2)}_1, \ldots, \tilde{G}^{(2)}_Q$. We can assess reliability of these estimates at regional scale by computing the scores $Sc^{(11)}$, $Sc^{(22)}$, $Sc^{(12)}$ and

$Sc^{(21)}$ of Section 3.1.2, where $\hat{G}_i$ is replaced by $\tilde{G}_i$. Note all these scores are cross-validation scores since the estimates $\tilde{G}^{(1)}_1$ and $\tilde{G}^{(2)}_1$ are computed without using any data of station $i$.

Second, we consider the set of all $\hat{\theta}^{(1)}_{i,j}$, $i = 1, \ldots, Q$ to estimate the surface response $\tilde{\theta}^{(1)}_j(l)$. We store the value of this function at every station location, giving new estimates $\tilde{G}^{*(1)}_1, \ldots, \tilde{G}^{*(1)}_Q$. Note that in the particular case of kriging, $\tilde{G}^{*(1)}_i$ is exactly $\hat{G}^{(1)}_i$ since it is an exact interpolation method, so every $\hat{\theta}^{(1)}_{i,j}$ equals $\tilde{\theta}^{*(1)}_{i,j}$. We can assess the stability of the interpolated

distributions at a given location $l_i$ when observations are available or not at this location by comparing $\tilde{G}^{*(1)}_i(r)$ and $\tilde{G}^{(1)}_i(r)$ for all $r$. For this we discretize $r$ between 0 and 450mm (which is the overall maximum rainfall) with 1mm step and we compute the total variation distance (TVD) between $\tilde{G}^{*(1)}_i$ and $\tilde{G}^{(1)}_i$ and the Kullback–Leibler divergence (KLD, Weijs et al., 2010) from $\tilde{G}^{*(1)}_i$ to $\tilde{G}^{(1)}_i$, which are given by:

$$\text{TVD}^{(1)}_i = \sup_r |\tilde{G}^{*(1)}_i(r) - \tilde{G}^{(1)}_i(r)|$$

$$\text{KLD}^{(1)}_i = \int_r \tilde{g}^{*(1)}_i(r) \log \frac{\tilde{g}^{*(1)}_i(r)}{\tilde{g}^{(1)}_i(r)} \, dr,$$

where e.g. $\hat{g}^{(1)}_i$ is the density function associated to $\tilde{G}^{(1)}_i$. Note that the KLD is not symmetric. Written as such, it can be interpreted as the amount of information lost when $\tilde{G}^{(1)}_i$ is used to approximate $\tilde{G}^{*(1)}_i$, so considering $\tilde{G}^{*(1)}_i$ as the "true"



| Class | Year | Winter | Spring | Summer | Autumn |
|-------|------|--------|--------|--------|--------|
| WP1 | 27% | 22% | 29% | 25% | 31% |
| WP2 | 36% | 47% | 35% | 28% | 34% |
| WP3 | 37% | 31% | 36% | 47% | 35% |

**Table 4.** Yearly and seasonal statistics of occurrence for the three WPs for the period 1948-2013. Winter extends from December to February, Spring from March to May, Summer from June to August, and Autumn from September to November.

distribution of data. TVD and KLD differ in that TVD focuses on the largest deviation between the two CDFs, whereas KLD somewhat integrate deviations along rainfalls. Obviously, one would like the interpolation to be as stable as possible when data are available or not at station $i$, i.e. that the lower $\text{TVD}_i$ and $\text{KLD}_i$, the more stable the interpolation at station $i$.

Regional scores $\text{MEAN}(\text{TVD}_i^{(1)})$ and $\text{MEAN}(\text{KLD}_i^{(1)})$ are then obtained by computing the mean of the $Q$ values. $\text{MEAN}(\text{TVD}_i^{(2)})$

and $\text{MEAN}(\text{KLD}_i^{(2)})$ are obtained similarly for the subsample $C^{(2)}$. For competing models, the closer the means are to $0$, the more spatially stable is the interpolation. For shortness, we will refer to $\text{MEAN}(\text{TVD})$ and $\text{MEAN}(\text{KLD})$ as the TVD and KLD scores, respectively (Table 2).

### 3.3    Procedure of model selection at regional scale

We wish to evaluate and compare performance of both marginal and mapping models for estimating rainfall frequency across

the region. We consider models both with and without season/WPs. For the cases involving the use of WPs, we use the WP classification described in Garavaglia et al. (2010). However a grouping of the eight WPs into three is made in order to improve the robustness of the method while conserving the diversity of the rainy synoptic situations. The choice of the grouped WPs is made in a separate analysis based on the spatial correlation of rainfall in the different WPs. The range of spatial correlation is twice as big in WP1 than in WP2, and three times as big in WP1 than in WP3. The occurrence statistics of the three WPs for

the period 1948-2013 are presented in Table 4. The three WPs spread relatively uniformly across the year, although WP2 tends to be more frequent in winter while WP3 is more frequent in summer.

In cases where subsampling is also undertaken by season, we impose a restriction of $S$ being two seasons, representing the season-at-risk during which most of the annual maxima are observed, and the season-not-at-risk. Furthermore, we impose the season-at-risk to be the same for all the stations due to the little extent of the region, defining the season-at-risk as the three

months of September, October and November, as in Garavaglia et al. (2010); Evin et al. (2016) for example. Alternative for bigger regions would be to select the months composing the season-at-risk following the procedure described in Blanchet et al. (2015).

### 3.3.1    Marginal selection procedure.

First we consider the marginal distributions of Table 1 and select the best of them at regional scale, as described in Section

25   3.1.2:





1. We randomly divide the days of 1948-2013 into two subsamples of equal size, denoted $C^{(1)}$ and $C^{(2)}$.

2. For every station $i$, we consider the set of observed days in $C^{(1)}$ and $C^{(2)}$, giving $C_i^{(1)}$ and $C_i^{(2)}$.

3. We fit each distribution of Table 1 to the two subsamples, getting estimates $\hat{G}_i^{(1)}$ and $\hat{G}_i^{(2)}$ of each distribution and for every station.

4. We compute the scores of Section 3.1.2, getting two calibration scores ((11) and (22)) of NRMSE, $FF$ and $N_T$ and two cross-validation scores ((12) and (21)) of NRMSE, $FF$, $N_T$ and SPAN$_T$. For $N_T$, we consider $T = 5$ years, which is lower than the minimum length of the calibration data and allows focusing on the tail but still having several exceedances of the $T$-year return level at every station. So $FF$, by focusing on the maximum of roughly 10 to 30 years of data, can be seen as an evaluation score for the far tail, while $N_5$ can be seen as a evaluation score for the close tail. For SPAN$_T$,

we consider $T = 100$ and $T = 1000$ years in order to test extrapolation far in the tail but at a scale still commonly used for engineering purposes (dam building, protections etc, Paquet et al., 2013).

5. We repeat 50 times the steps 1-4.

We obtain 100 values of each calibration score and 100 values of each cross-validation score. We apply this procedure to the four distributions of Table 1, considering the four alternatives: no season nor WP ($S = 1$, $K = 1$), two seasons but no

WP ($S = 2$, $K = 1$), no season but three WPs ($S = 1$, $K = 3$), two seasons and three WPs ($S = 2$, $K = 3$). Comparing the distributions of the scores of the 16 models allows us the select the marginal distribution yielding to the best fit at regional scale. We select this marginal model for further consideration.

### 3.3.2  Mapping selection procedure.

Second we consider the mapping models of Section 3.2.1 for interpolating the selected marginal model, and we select the best

of them in two ways, as described in Section 3.2.2.

1. We consider the estimates $\hat{G}_i^{(1,t)}$, $i = 1, \ldots, Q$, obtained at the $t$-th iteration of the marginal selection procedure, and corresponding to the subsamples $C_i^{(1,t)}$, $i = 1, \ldots, Q$.

2. We estimate the mapping models of Section 3.2.1 following the leave-one-out cross-validation framework of Section 3.2.2. We obtain new estimates $\tilde{G}_i^{(1,t)}$ for each station $i$ and each mapping model. Each $\tilde{G}_i^{(1,t)}$ is a cross-validation

estimation of both $G_i^{(1)}$ and $G_i^{(2)}$ since the computation of $\tilde{G}_i^{(1,t)}$ did not use any data of station $i$.

3. We compute the scores of Section 3.1.2 associated to $\tilde{G}_i^{(1,t)}$, $i = 1, \ldots, Q$. We obtain for each score two values (e.g. $FF^{(11)}$ and $FF^{(21)}$ when considering $\tilde{G}_i^{(1,t)}$ and the maximum value over either $C^{(1)}$ or $C^{(2)}$). All these score are cross-validation scores.

4. We estimate the mapping models of Section 3.2.1, using all the stations to make interpolation. We obtain new estimates

$\tilde{G}_i^{*(1,t)}$ for each station $i$ and each mapping model.





5. We compute the spatial means of the TVD and KLD scores of Section 3.2.2, comparing $\tilde{G}_i^{*(1,t)}$ to $\tilde{G}_i^{(1,t)}$, for , $i = 1, \ldots, Q$.

6. We repeat steps 1-5 for the estimates $\tilde{G}_i^{(2,t)}$ corresponding to the subsample $C_i^{(2,t)}$.

7. We repeat steps 1-6 for each of the 50 subsamples.

We obtain 200 values of each cross-validation score NRMSE, $FF$, $N_T$ and SPAN, and 100 values of the TVD and KLD scores. Comparing the distributions of these scores allows us the select the mapping model yielding the smallest score, for the selected marginal model. We select this mapping model for further consideration.

At this step we have selected the best marginal model and the best mapping model (among those tested) for our data.

### 3.3.3  Final regional model.

Finally, we consider the whole sample of data and apply the selected marginal distribution and mapping model:

1. We estimate the selected marginal distribution $\hat{G}_i^*$ based on the full data, giving parameters $\hat{\theta}_{ij}^*$, $i = 1, \ldots, Q$.

2. We estimate the mapping model associated to each marginal parameter, using all $\hat{\theta}_{ij}^*$, $i = 1, \ldots, Q$, to estimate the surface response $\tilde{\theta}_j^*(l)$.

We obtain estimates of $\mathrm{pr}(R(l) \le r | R(l) > 0)$ for every $l$ within the region, making full use of the observations. Estimation
of $\mathrm{pr}(R(l) \le r)$ is obtained straightforwardly from (3). Although not considered in this study, confidence intervals could be obtained by bootstrapping within these two last steps.

## 4  Results

### 4.1  Selection of the marginal distribution

We show in Figure 4 the influence of considering seasons and/or WPs in the marginal distributions, in the case of the Gamma
distribution for illustration, but similar patterns are found with the other distributions. Figure 4 depicts the cross-validation scores of NRMSE, $FF$ and $N_5$ and the reliability score SPAN$_{100}$ for the 100 split samples $C^{(1)}$ and $C^{(2)}$. Calibration scores are not shown because they are very similar to the cross-validation scores (correlation 91% between validation and calibration scores). For the stability criteria, we only show the values of SPAN$_{100}$, which corresponds to 3 to 10 times the length of calibration data, but actually values for $T = 1000$ years lead to the same conclusions (correlation 99.9% between SPAN$_{100}$ and
SPAN$_{1000}$).

Comparing the reliability scores NRMSE, $FF$ and $N_5$ when neither season nor WP is used (case (1,1)) with cases when either WPs (case (1,3)) or seasons (case (2,1)) are considered shows there is at regional scale a clear improvement of using a mixture of Gamma distributions rather than considering a single Gamma for the whole year. Reliability criteria are slightly better (i.e. lower) when WPs are considered rather than season, but this is more marked for the bulk of the distribution (represented by the

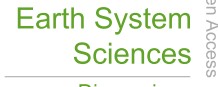
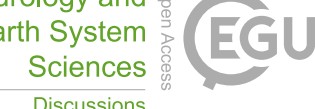

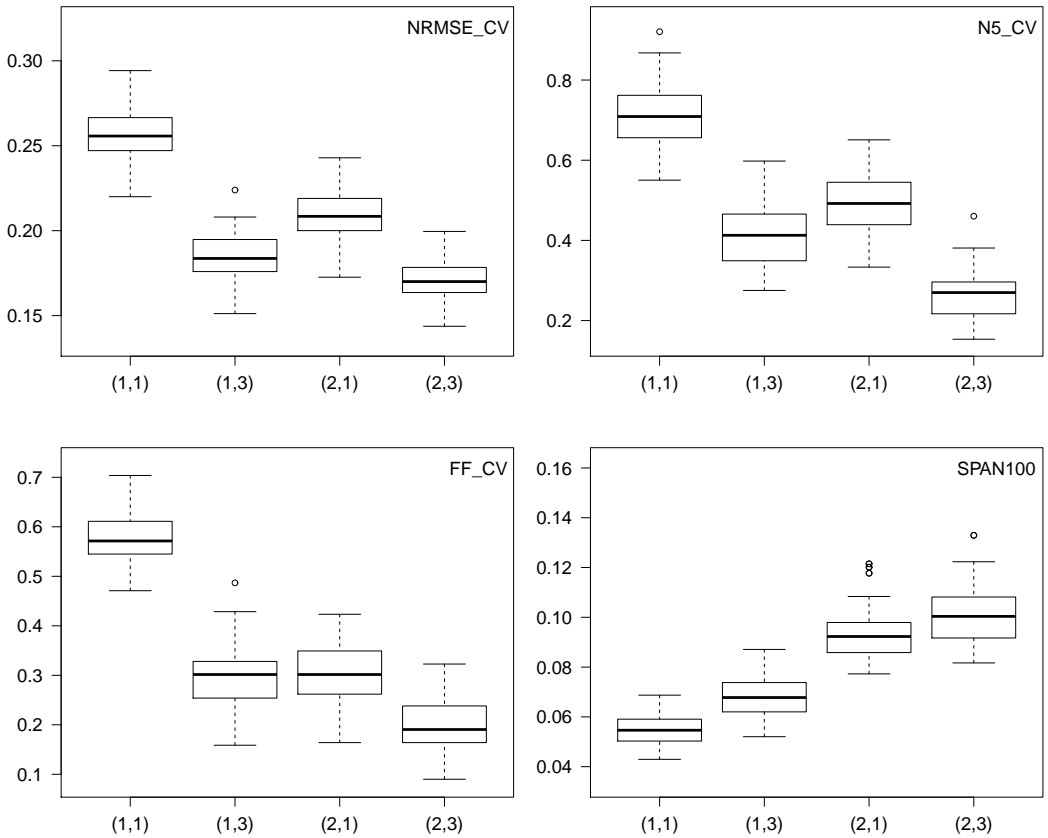

**Figure 4.** Scores of cross-validation when $G_{s,k}$ are Gamma distributions and the number of seasons and WT varies: $S \in \{1, 2\}$ and $K \in \{1, 3\}$. The values of $(S, K)$ are indicated in the x-labels. Each boxplot contains 100 points.

NRMSE scores) than for its tail ($FF$ and $N_5$). Reliability scores are even better when both seasons and WPs are considered (case (2,3)), particularly for the tail of the distribution.

   Obviously, there is a loss of stability when considering seasons and/or WPs due to the increased number of parameters. However the score of SPAN$_{100}$ ranges 0.08-0.14, which means that the two estimates of the 100-year return levels over $C^{(1)}$

5  and $C^{(2)}$ differ by 8 to 14%, which seems acceptable.

   We illustrate the quality of the fit for the station Antraigues, located in the very foothills of the Massif Central slope (see Figure 1). We focus on the tail of the distribution by looking at the return level plot (here beyond the yearly return period). Antraigues is chosen to illustrate the case of an heavy-tailed distribution of rainfall: its 20-year return level is about 3 times larger than its yearly return level. Of course, some variability is found in the return level estimations depending on the sub-

10  sample used for estimation. Figure 5 illustrates this by showing the 95%-envelope of return level estimations over the 100 subsamples on either $C^{(1)}$ or $C^{(2)}$ together with the full sample of 35 years. Note that the envelopes do not show confidence



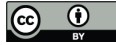

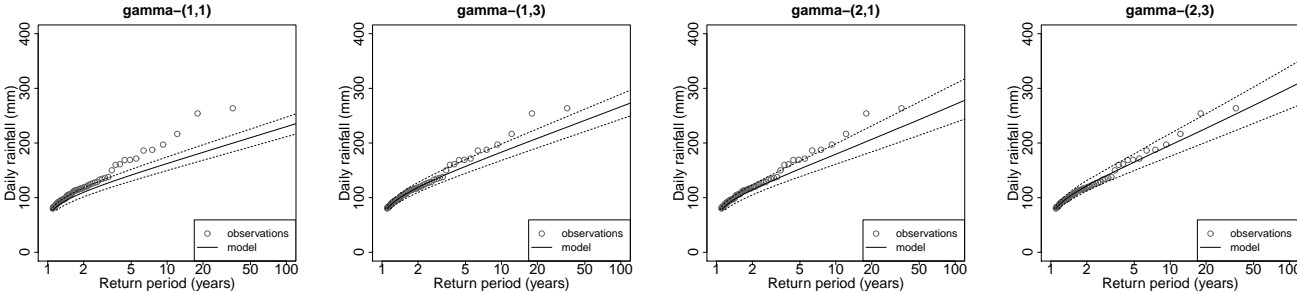

**Figure 5.** Case of Antraigues when $G_{s,k}$ are Gamma distributions and the number of seasons and WT varies: $S \in \{1, 2\}$ and $K \in \{1, 3\}$. The values of $(S, K)$ are indicated in the title. The dotted lines show the 95%-envelope of return level estimates over the 100 subsamples. The plain line shows the median estimates. The gray points show the full sample (35 years). Each estimation is based on half of these points.

intervals (that could be obtained by bootstrapping for example) but variability when only half the data in used from calibration. Figure 5 clearly shows that considering seasons or WPs allows to get heavier-tailed distributions. The median estimate follows quite closely the empirical points. Variability is relatively low, altough it is slightly larger than with the other marginal models involving less parameters. Coefficient of variation of the 100-year return level is less than 7%, in coherence with the SPAN$_{100}$

of Figure 4 at regional scale.

Due to its better fit for the Gamma model (Figures 4 and 5) as for the other distributions (not shown), we select the mixture model with $S = 2$ seasons and $K = 3$ WPs for further investigation. Figure 6 shows the scores of cross-validation when the parent distribution $G_{s,k}$ is either the extended exponential, extended Generalized Pareto, Gamma, lognormal or Weibull distribution. The reliability scores NRMSE, $FF$ and $N_5$ in the lognormal case are missing because they lie far above the upper

range of the depicted values (e.g. the median NRMSE is about 0.7), which clearly rules out the use of the lognormal model for this region. The reliability criteria of the four other distributions all show the same pattern: a better performance of the Gamma model, closely followed by the extended exponential case. Then comes the extended Generalized Pareto, itself closely followed by the Weibull model. A closer look at the values of $ff_i$ and $n_{i,5}$ for all stations and samples reveals that the weaker reliability of the Weibull and extended Generalized Pareto models is due to their tendency to systematically overestimate the probability

of occurrence of large values (i.e. to underestimate their return period), with $ff_i$ and $n_{i,5}$ tending to be too frequently small (remind case $G_1$ of Figure 3).

The stability score SPAN$_{100}$ in Figure 6 shows that the most stable model is the lognormal case but this is because the lognormal distribution gives unreasonably huge estimates of large return values (as illustrated in Figure 7 for the station Antraigues, for example), giving very large normalization terms in the SPAN criteria (see (8)). The fact that the lognormal

model has by far the worst reliability scores but the best stability score preaches for the conjoint use of these two family of scores not to misinterpret results. Stability of the Gamma and extended exponential distributions are very similar and fairly less good than the lognormal case. Then comes the Weibull distribution, and finally the Generalized Pareto distribution, which is clearly the least stable.




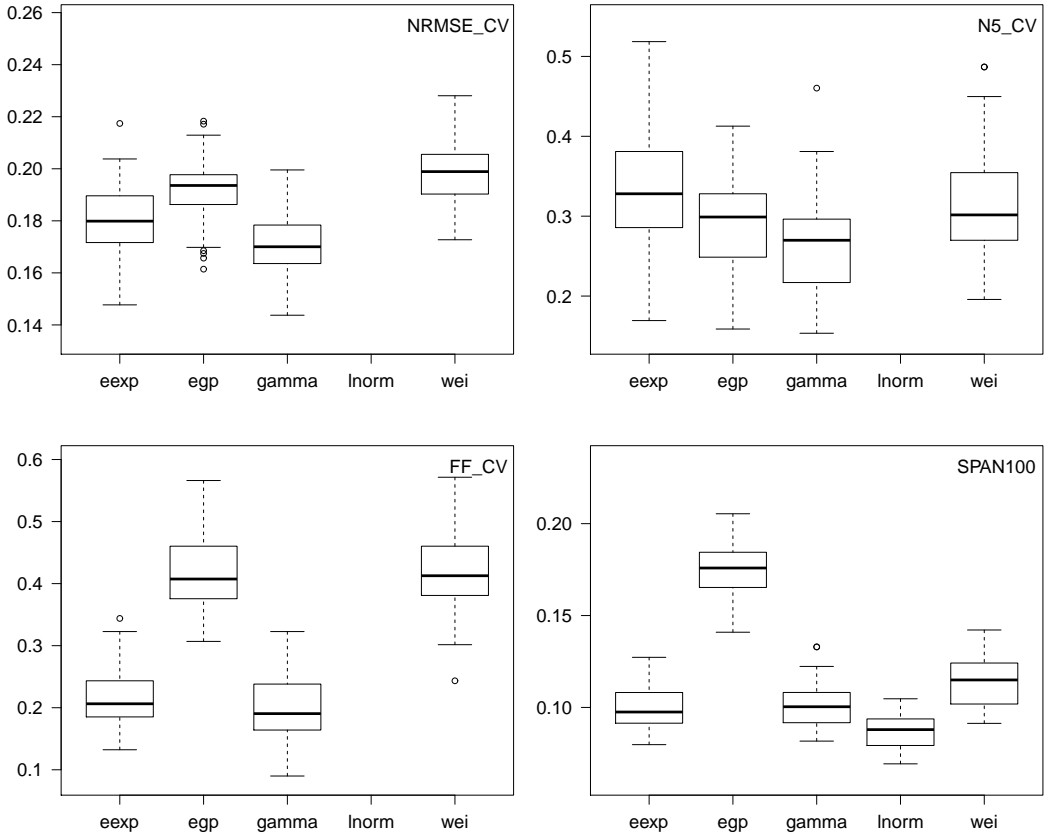

**Figure 6.** Scores of cross-validation when $G_{s,k}$ is either the extended exponential (eexp), extended Generalized Pareto (egp), Gamma (gamma), lognormal (lnorm) or Weibull (wei) distribution, with $S = 2$ and $K = 3$. Each boxplot contains 100 points. The boxplots of reliability scores in the lognormal case are missing because they lie far above the upper range of depicted values.

Figure 7 illustrates the quality and spread of the fits depending on the distribution for the station Antraigues, when estimation is made on either subsample. Compared to Figure 5, it confirms that the Gamma and the extended exponential models perform almost likely. Median estimations differ by about 5% for the 100-year return level (303mm for the Gamma vs. 287mm for the extended exponential model) and by about 7% for the 1000-year return level (414mm vs. 386mm), with very similar widths of the 95% envelopes (e.g. $\pm$ 40mm for the 100-year return level). The lognormal model clearly fails to reproduce return periods larger than one year, giving much too heavy tails despite a resonably good fit of the bulk. Actually the skewness –which informs somehow on the "asymmetry of the bulk"– is reasonably well estimated, whereas the kurtosis –which informs on the heaviness of the tail– is much overestimated. This is in line with Fig. 2 of Hanson and Vogel (2008), which shows that when the skewness of daily rainfall across the US is well estimated by the lognormal distribution, then the kurtosis is much overestimated. Note that Papalexiou et al. (2013) did not find such ill-fitted tails with the lognormal distribution but in their case fitting is made




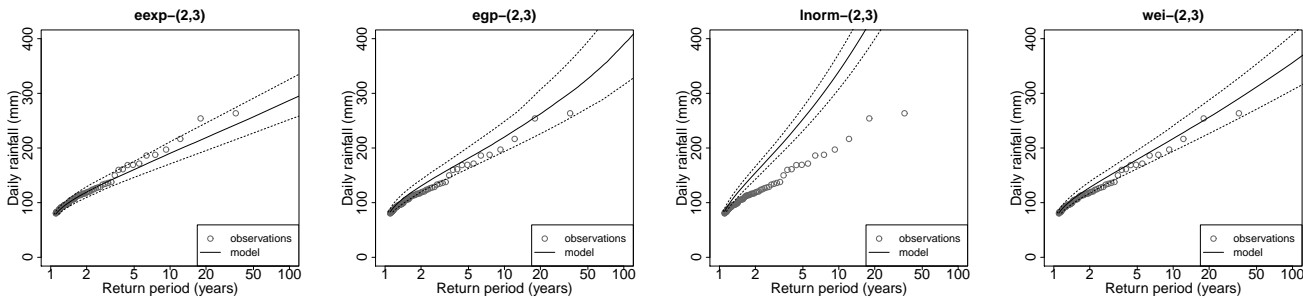

**Figure 7.** Case of Antraigues when $G_{s,k}$ is either the extended exponential (eexp), extended Generalized Pareto (egp), lognormal (lnorm) or Weibull (wei) distribution, with $S = 2$ and $K = 3$. The dotted lines show the 95%-envelope of return level estimates over the 100 subsamples. The plain line shows the median estimates. The gray points show the full sample (35 years). Each estimation is based on half these points. Case of the Gamma distribution is shown in the right panel of Figure 5.

on the tail (i.e. on the largest values), whereas the lognormal model seems to fail when adjusting both the bulk and the tail of rainfall distribution. The Weibull and extended Generalized Pareto models give very similar fits up to the 50-year return period but the return level plot of the extended Generalized Pareto model is more convex than for the Weibull model, giving median estimation 8% larger for the 100-year return level (390mm vs. 358mm) and 35% larger for the 1000-year return level (799mm

vs. 522mm). Width of the 95%-envelope is also larger both in absolute and relative values, in coherence with the SPAN$_{100}$ of Figure 6 at regional scale. Finally both the Weibull and extended Generalized Pareto models overestimate the return levels associated to 1-5 years, unlike the Gamma and extended exponential models. This tendency towards overestimation of the tail is actually a quite general feature observed for most the stations, giving to frequently low values of $ff_i$ and $n_{i,T}$, as already stated.

The results of Figures 6 and 7 lead us to conclude that the best performance for the region is achieved by the Gamma and extended exponential models, which actually perform very similarly for Antraigues station. Due to its slightly better performance at regional scale for adjusting the tail of the distribution ($FF$ and $N_5$ in Figure 6), we select the Gamma model (with two season and three WPs) for further consideration.

## 4.2   Selection of the mapping model

Figure 8 shows the 10 scores of evaluation of the mapping models of Table 3. The first comment is that, compared to Figure 6, the $FF$ scores of leave-one-out cross-validation are for any mapping method of the same order as for the the local fits, while the SPAN scores are even slightly better. This means that i) no mapping method gives systematic over- or under-estimation of the very tail, and ii) mapping gives more stable estimations by smoothing out the sampling effect. However NMRSE scores are all larger, meaning that any mapping gives less accurate estimations of the full distributions than the local fits. Loss in accuracy

is equivalent and relatively small for all kriging interpolations and the bivariate thin plate splines (with or without drift), while the trivariate thin plate spline and even more the linear model are less accurate. A closer look at the fits of all stations reveals





that the strong loss in NRMSE for these two methods is actually due to few stations that are systematically very badly fitted -among which the station of Mayres of Figure 9-, which strongly deteriorates the spatial mean of the scores. Their less good performance is due to a lack of flexibility, which prevents them from adapting to local features. However in the same time, the lack of flexibility of these methods allow them for slightly increased stability in the tail, as shown by the SPAN scores in

Figure 8.

Back to the kriging methods, the three tested alternatives give very similar fits, with slightly less stability when considering a drift in station altitude $z$, while considering the smoothed altitude $Z$ is useless because $a_1$ in (11) is almost always zero. The best kriging method for our region study in thus the simple kriging interpolation. This method is only slightly beaten in accuracy by the bivariate thin plate spline (with or without drift), but which is slightly less stable. However the TVD and

KLD scores comparing the spatial stability of the mappings show that the bivariate thin plate splines are clearly more stable in space than all kriging methods. The linear model is even more stable but, as already said, it is much less accurate. Finally, comparing the five cases of thin plate spline shows that the three bivariate cases clearly outperform the trivariate case, both in terms of accuracy and stability. Comparing the bivariate case with drift (14) to the the trivariate case (15) shows the usefulness of considering non-linear weights of the distance (through the term $h_i^2 \log(h_i)$ vs. $h_i'$). Last but not least, whatever the method

but particularly for the thin plate spline, better accuracy and stability is achieved when the smoothed altitude $Z$ is considered rather than the station altitude $z$, as also found in Hutchinson (1998) for interpolating rainfall data.

We illustrate the results in Figure 9 for the Antraigues station, adding to that the case of the worst fit of the thin plate spline, which is for the station of Mayres. Mayres lies at about 500 m.a.s.l, as Antraigues, but it is located at the end of a funnel shaped valley surrounded by steep slopes (see Figure 1). This creates favorable conditions to the orographic intensification of rainfall,

with the consequence that Mayres receives more rainfall than expected at this altitude, as also confirmed by Figure 2. For this reason, although the local fit of the Gamma model is reasonably good, the interpolated distributions underestimate the empirical values, even the small ones. This can be seen in Figure 9 by comparing the black curves, which were obtained independently on the data of Mayres, to the red curves of the kriging case, which are equal to the local fits since kriging is an exact interpolation. Although return levels are underestimated with all models, kriging and the bivariate thin plate spline manage however better

the fit the data of Mayres in the leave-one-out framework, in coherence with the NMRSE values of Figure 8 at regional scale. For the station Antraigues, underestimation is also found for all methods due to smoothing but with much smaller extent than for Mayres. For both stations, comparing the red and black curves shows that kriging and the trivariate thin plate spline are too dependent on the data used for fitting since large differences are obtained whether the station is included or not in the estimation, in coherence with the TVD and KLD values of Figure 8 at regional scale. Finally, comparing the envelope widths

in red and black in Figure 9 confirms that interpolation increases stability of the estimates, as also revealed by the SPAN score of Figure 9.

We conclude following the results of Figures 8 and 9 that the best interpolation method (among those tested) is the bivariate thin plate spline with drift in smoothed altitude, which is slightly more accurate but much more stable spatially stable than the kriging method. The trivarite thin plate spline and the linear model should be avoided for our data due to their lack of flexibility.





**Figure 8.** Scores of mapping when $G_{s,k}$ are Gamma distributions with $S = 2$ and $K = 3$ whose parameters are interpolated with the mapping models of Table 3. The two first rows show leave-one-out cross-validation scores. Each boxplot contains 200 points. The third row compares interpolations at a given station whether the data of this station are used or not in the interpolation. Each boxplot contains 100 points.





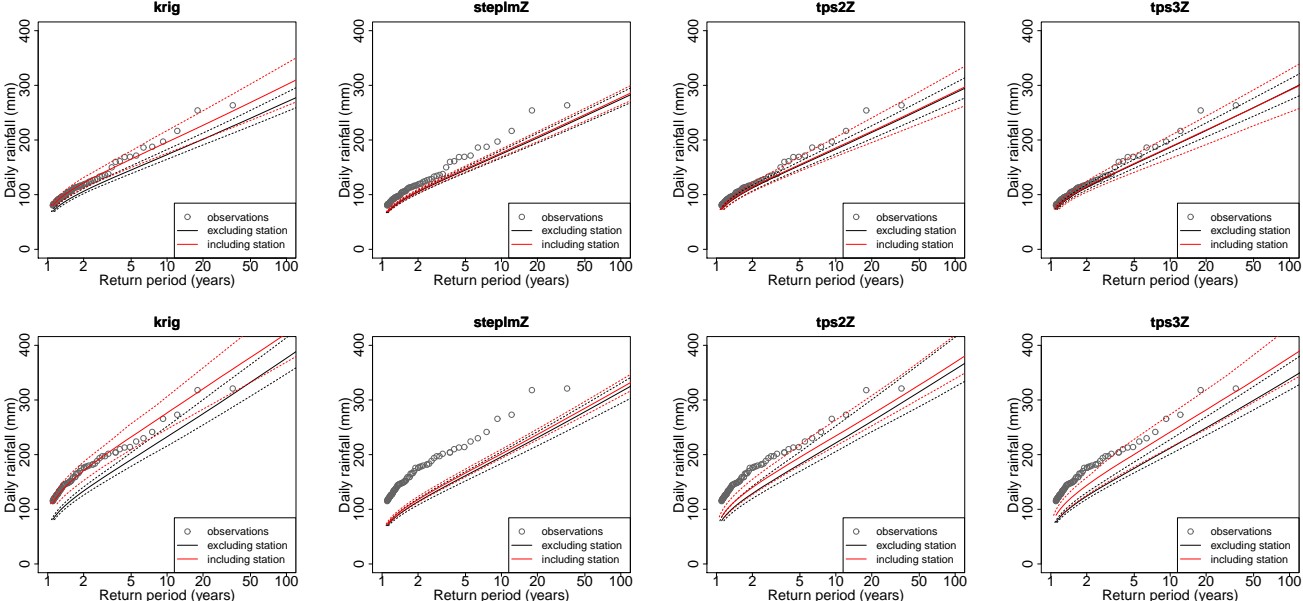

**Figure 9.** Case of Antraigues (top) and Mayres (bottom) when $G_{s,k}$ are Gamma distributions with $S = 2$ and $K = 3$ whose parameters are interpolated with either: kriging without extrenal drift (krig), stepwise linear model (steplmZ), bivariate thin plate spline with drift (tps2Z), or trivariate thin plate spline (tps3Z). The dotted lines show the 95%-envelope of return level estimates over the 100 subsamples. The plain line shows the median estimates. In black, each interpolation is based on half the data of the other stations, excluding the considered station. In red, interpolation is based on half the data of all the stations, including the considered station. The gray points show the full sample (35 years for both stations).

## 4.3 Final regional model

Figure 10 illustrates the final regional models when both the Gamma parameters and the mapping models are estimated using all the available data. The map of the probability of daily rainfall to exceed 1mm is obtained from (3) with $r = 1$mm. The maps of the mean nonzero rainfall in the WPs of the season-at-risk ($S2$) are obtained as the product $\lambda_{2,k}\kappa_{2,k}$, $k = 1, \ldots, 3$, with the
5  notations of Table 1. The four maps of Figure 10 reveal the double effect of the Massif Central ridge, which both creates a climatological border and enhances orographic precipitation. The map of rainfall probability is conform to the climatology of the region (as shown by the colored points), with a smaller probability of rainfall in the Rhône valley and increased probability when approaching the relief due to orographic effect. Being more exposed to the west fluxes -which are the most common in the region-, the west side of the Massif Central undergoes more frequent rainfall events. Comparing the three maps of mean
10  nonzero rainfall reveals very different range of values depending on the WP, with WP1 showing much larger values than the other WPs all across the region. Recall that the WPs were constructed based on the spatial correlation of rainfall, with WP1 showing a spatial correlation of rainfall twice as big as in WP2 and three times as big than in WP3. Remarkably, roughly the same factors are found when comparing the range of values of the means (resp. 5-36mm, 3-11mm, 2-10mm). There is thus a



strong link between the spatial correlation of rainfall and the mean amounts. However the WPs do not only differ in the range of values of the mean amounts but also, and maybe even more, in the way these amounts are usually distributed over the region. This emphasizes once again the usefulness of considering subsampling over WPs in order to distinguish contrasted spatial pattern. The map of WP1 shows a strong intensification of rainfall along the Massif Central slope, while a clear decrease in

the mean rainfall is found when passing the Massif Central ridge both towards the Massif Central plateau with means divided by three in 20km, and towards the Rhône valley with means divided by two in 20km. In WP2 the topography builds somehow a mask effect. The larger means are found along the Massif Central slope with a fast break when passing the Massif Central ridge. Daily means in the Massif Central plateau are half the values of the slope, while daily means in the Rhône valley are just slightly lower than in the slope. Finally the map of the mean nonzero rainfall in WP3 shows an inverse pattern to that of

the probability of rainfall. The mean almost linearly decreases from the Rhône valley to the Massif Central plateau while the probability of rainfall almost linearly increases. The largest rainfall events in this WP are usually convective events of small extent occurring mainly in the Rhône valley, reason why the mean values are larger in this area, although the probability of rainfall is relatively low.

Last but not least, Figure 11 shows the map of the probability of daily rainfall to exceed 100mm. It reveals a clear concen-

tration of higher probabilities of exceedance along the Massif Central ridge, with actually quite similar patterns as the averages of annual totals and annual maxima in Figure 2, with however much more pronounced disparities. It is up to 10 times less likely to exceed 100mm rainfall in the Rhône valley than along the ridge, and up to 1000 times less likely in the Massif Central plateau. Actually 100mm is expected to be exceeded several times a year along the ridge, about every year on the slope, and on average every 100 to 1000 year in the Massif Central plateau.

## 5   Conclusion and discussion

In this article we have presented an objective framework for estimating rainfall cumulative distribution function within a region when data are only available at rain gauges. For this we have proposed an objective procedure involving split sampling cross-validation and using several evaluation scores allowing to validate the accuracy of both the bulk and tail of the distribution and the stability of these estimates when calibration data changes. We have applied this procedure to daily rainfall in the

Ardèche catchment in southern France, a particularly challenging test case showing very strong disparities in rainfall in very short distance. Our results show that for this region, the best marginal model (among those tested) is a mixture of Gamma distributions over seasons and weather patterns, and that the best mapping model (among those tested) is the bivariate thin plate spline.

Although our procedure of selection is general and could be applied to any region of the world -and possibly to other variables

(temperature etc)-, we stress that our conclusions are in themselves not universal. In particular, other marginal distributions may be more suitable than the Gamma in other regions of the world showing less or more heavy tails. Although the mixture of distributions over weather patterns has revealed efficient in other countries (e.g. in Norway and Canada, Brigode et al., 2014; Blanchet et al., 2015), it might be less relevant in, e.g., monsoon climate regions where the consideration of seasons seems





**Figure 10.** Map of the probability of daily rainfall to exceed 1mm and of the mean of nonzero rainfall in the three WPs of the season-at-risk. The points are colored with respect to the empirical estimates.





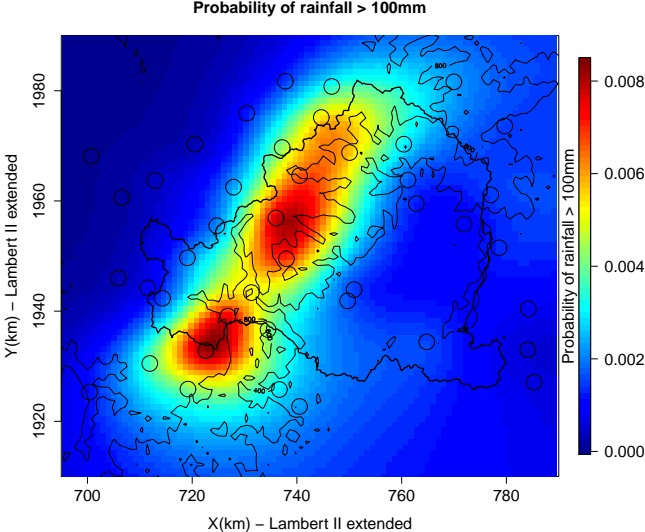

**Figure 11.** Map of the probability of daily rainfall to exceed 100mm. The points show the location of the stations.

essential and might be enough. We also stress that the goal of this article was neither to be exhaustive nor to find *the* best model for the region, but rather to present an objective cross-validation framework that can be applied to other variables and/or models of interest. Possible direction for improvement regards the choice of the spatial covariates to be used in the interpolation method. There might be more relevant covariates than the geographical coordinates used in this study, e.g. considering atmospheric and

terrain characteristics (Carreau et al., 2013; Kyriakidis et al., 2001). Finding good gridded covariates (and good regression models) is a subject of research in itself, and it is particularly tricky in areas with complex topography (Prudhomme and Reed, 1999; Weisse and Bois, 2001; Drogue et al., 2002; Beguería et al., 2009; Rogelis and Werner, 2013). The geographical distance itself might also be improved, e.g. by better accounting for the terrain characteristics (Gottardi et al., 2012; Evin et al., 2016) or by considering statistical distance (Ahrens, 2006). Also more robust estimates of the marginal parameters at station locations

(i.e. of the $\hat{\theta}$s) might be obtained by gathering observations of neighbor stations in order to increase the sample size, following the concept of regions-of-influence proposed by Burn (1990). Such idea has been quite widely used in the context of rainfall extremes (e.g. Carreau et al., 2013; Evin et al., 2016, for the studied region). However we anticipate the gain to be much less pregnant when interest is in modeling *any* rainfall –as in this study–, and not only the extreme ones since parameter estimation is already based on many data (several thousands).

Despite the above reservations of prudence, some other results seem to us to be generalizable. Among these is the fact that distributions showing sub-exponential tails (EGP for example) give usually unrealistic return levels when extrapolating far out the observations. Such distributions should not be used if return periods of several hundreds of years are of interest, as it is in many cases related to civil protection (dykes, dams etc, Paquet et al., 2013). Also the kriging method gives usually too patchy maps of rainfall hazard by sticking the observations, unless nugget effects are considered (which was not the case in this study).



Finally the linear model with spatial covariates usually fails to map rainfall hazard because it is highly unlikely to be ruled by simple enough physics for the parameters to be well represented as linear functions of the covariates, in particular in such complex topography (Carreau et al., 2013).

Last for not least, we put this study in a framework of temporal stationarity and we addressed the question of the spatial

nonstationarity of the margins. Yet several studies showed temporal trend in the rainfall distribution in the region, particularly since the 80s and particularly along the Massif Central slope where daily rainfall is usually more intense (Blanchet et al., 2016b; Tramblay et al., 2011, 2013; Vautard et al., 2015). Extending the proposed procedure to the case of nonstationary rainfall would be possible by considering the marginal parameters as parametric functions of time, e.g. linear models. This would increase the number of parameters but the split sample framework would still be valid. However the scores would have

to be adapted to account for changing distributions. One way of doing this would be to transform the rainfall at time $t$ to some variate independent of $t$. For example considering $R'_t = \exp\{-\exp(-G_t(R_t))\}$ would transform $R_t$ with CDF $G_t$ to a stationary Gumbel variate, to which the scores presented in this article could be applied for model evaluation and selection. A drawback however would be that the value of the scores would depend upon the chosen transformation. Also the SPAN score might have to be thought over because return levels in changing climates are not meaningful for quantifying risk (Katz, 2013).

*Author contributions.*  J. Blanchet developed the cross-validation framework, wrote the code in R (R Core Team, 2013) and prepared the manuscript. The estimation of the margins is partly based on a code written by P. Vaittinada. The climatological discussion of the produced hazard maps has benefited from the input of E. Paquet and D. Penot.




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
