# Peer review of "Mapping rainfall hazard based on rain gauge data: An objective cross-validation framework for model selection"

_Hydrology and Earth System Sciences, 2018_

## Referee Comment (RC1) · R. Katz (Referee) · 26 Apr 2018

**GENERAL COMMENTS**

This paper deals with the statistical modeling of the distribution of rainfall amounts within a region, especially focusing on extremes. The approach is computationally intensive, with the parameters of the rainfall distribution at individual sites being mapped across the region using (e.g.) Kriging or splines. Cross validation is applied to evaluate the performance of the candidate models (e.g., form of distribution and spatial interpolation technique). Challenges include the attempt to model the entire range of rainfall amounts (i.e., from near zero to the most extreme) with a single distribution.

[Figure]

It is claimed that the proposed approach to rainfall modeling is "general and could be applied to any region of the world" (p. 24, line 29). Yet some aspects of the approach seem tailored to the application to a specific region in France. In particular, seasonality is treated by dividing the year into two seasons, one in which extremes typically occur. Plus the model is fitted conditional on one of three possible weather patterns (WPs), based on the spatial correlation of rainfall for the region. Although the number of seasons and WPs could certainly be varied to model rainfall for other types of climate, it is not clear that the constraint of being limited to a quite small number of seasons and WPs could always permit an adequate fit.

Another potential limitation concerns the performance of the different forms of distributions fitted to rainfall amounts, particularly for extremes. Conclusions are drawn about "heavy tails" that could benefit by relying more on extreme value theory. The restriction to a single distribution may have distorted the performance for extremes, with some of the conclusions conflicting with results in the literature when only extremes are modeled.

For these reasons, I recommend that the manuscript be accepted for publication subject to revision.

SPECIFIC COMMENTS

(1) Generality of proposed approach

It seems like a crude approximation to consider only two seasons and assume stationarity within a given season. More realistic approaches include allowing the parameters of the rainfall distribution to gradually change depending on the time of year. Some regions of the world even have more than one wet season, indicating a limitation of the proposed approach.

Conditioning on a few WPs based on the degree of spatial correlation of rainfall is an intriguing and not very common approach. Alternatives in the literature have included either introducing a hidden state variable (likewise assuming only a few possible states), which would require much more involved calculations, or including an observed covariate (such as an index of atmospheric circulation), which would require its identification for a given region but could assume effectively infinitely many possible states. Other than convenience, the advantages of the proposed approach are not clear.

(2) Using extreme value theory to interpret results

It is concluded that a mixture of gamma distributions provides the best fit, especially for extreme high precipitation amounts. Yet a gamma distribution has a light tail, well known to not be heavy enough for precipitation extremes. Still it is argued that allowing the gamma distribution to vary depending on the season and on the WP induces a heavier tail (Figure 5).

If this claim were correct for seasonality, then it would appear that the apparent heavy tail is at least partly an artifact of ignoring seasonality. Yet there is some evidence in the literature (e.g., by explicity allowing some of the parameters in an extreme value analysis to vary within the year) that this is not necessarily the case.

Concerning conditioning on WPs, it is well known that a mixture of gamma distributions can induce a heavier tail than a single gamma. Yet I wonder whether a mixture involving only a few gamma distributions (i.e., only three for the wet season) is sufficient to produce a truly heavy tail (in the sense of extreme value theory).

So it may be informative to examine how well the gamma distribution (and the mixture of three gamma distributions depending on the WP) fits precipitation amounts in the wet season alone. As it stands, I worry that the results for extremes may have been distorted by the constraint of fitting a single distribution to all rainfall amounts.

(3) Assumption of temporal independence

It is effectively assumed that the rainfall amounts at an individual site, especially extreme high values, are temporally independent (e.g., second displayed equation on p.

8 and p. 9, line 7). But this assumption never appears to be explicitly stated or verified.

There is some evidence in the literature of "clustering" at high levels for time series of daily rainfall amounts at individual sites. Cross validation, depending on how it is implemented, would not properly account for the effects of such temporal dependence.

EDITORIAL COMMENTS

(1) p. 3, lines 16-17

Not clear how the "factor" is defined or calculated.

(2) p. 25, Figure 10

Three of the graphs are for the same quantity, mean of non-zero rainfall for different weather patterns. But the color coding varies making comparisons difficult.

---

## Referee Comment (RC2) · Anonymous Referee #2 · 6 Jun 2018

The paper describes a method for mapping distribution parameters from single rain gauge record across a domain, but lacks a proper discussion of the role/significance of the proposed framework in the landscape of rainfall hazard mapping: in that respect, if the proposed framework is really a (significant) step forward then the authors should demonstrate it by comparing it to the state-of-the-art in rainfall hazard mapping, which they as such also describe in the introduction (page 2, line 4-14). I am left with the impression of reading a technical report rather than a paper significantly advancing the field (which does not mean there is no advancement per se, but it is difficult to judge at this stage).

[Figure]

I do not understand why classic interpolation of rainfall of a certain frequency comes along with issues of zero values (even more obvious when mapping amounts and their exceedance probabilities), see page 2, line 10-11. A certain exceedance probability of rainfall is >0 by definition, and we are talking about regional hazard maps and not about scenarios (i.e. rainstorms). Or am I getting something wrong? Also, provided the issue exists, the authors address a solution themselves (which is the analytical transformation), which brings me back to the issue of ideally comparing the proposed framework to the state-of-the-art.

I also do not understand why one would feed a distributed rainfall-runoff-model with a rainfall-frequency map (page 1, line 17-18). The resulting rainfall-runoff-model output is highly artificial, not much telling about a realistic hydrological scenario. In hydrological hazard/risk assessment, one would probably conduct scenario-based analyses based on potential (realistic) rainstorms or continuously simulate rainfall time series to feed an RR-model, to get insights about (extreme) runoff events. But maybe I did not understand what the authors intend to say here; citations of related work would maybe clarify.

Modelling only two seasons is a clear limitation, and so is the assumption of stationarity. The same applies to the three weather patterns, which is another constrain. I am also in doubt that the Gamma (or the mixture of a Gamma) is suitable in other regions, especially in the tail. That is, it remains open whether the framework is really applicable to other regions. The authors put that into question themselves (e.g. page 24, line 30, among others). So besides comparing the proposed method to the state-of-the-art, a second study area (other climate, more seasons) would be – in my eyes - very important.

What is also missing is a proper discussion of the uncertainties of the rainfall records in mountain regions. It would be important to consider these observation errors in the framework, again for the proposed model and the state-of-the-art in comparison, to really understand all implications.

A figure describing the entire framework would be important, it is difficult to follow all steps and practitioners will for sure appreciate a presentation that is a tad more "hands-on".

I would recommend a final proofread by a native speaker, there is quite a number of minor language related mistakes throughout the manuscript, not a big deal but just a few examples:

"One of the difficulty"; "models for nonzeros rainfall"; "Similar idea is used"; "in the same time"; "independently on each others"

Sharing the original observation data (other journals even demand it) to allow for reproducing the results is recommended.

In summary, at this stage, I am in doubt that the reader really understands the added value of the method, why and when established rainfall hazard mapping methods are competitive and if the method is applicable to another region/climate.

---

## Author Comment (AC1) · 3 Jul 2018

We warmly thank the Reviewer for his valuable comments. We provide below a detailed point-to-point reply to these comments. The proposed changes for the next version of the article are indicated in red.

*GENERAL COMMENTS*

*This paper deals with the statistical modeling of the distribution of rainfall amounts within a region, especially focusing on extremes. The approach is computationally intensive, with the parameters of the rainfall distribution at individual sites being mapped across the region using (e.g.) Kriging or splines. Cross validation is applied to evaluate the performance of the candidate models (e.g., form of distribution and spatial interpolation technique). Challenges include the attempt to model the entire range of rainfall amounts (i.e., from near zero to the most extreme) with a single distribution.*

*It is claimed that the proposed approach to rainfall modeling is "general and could be applied to any region of the world" (p. 24, line 29). Yet some aspects of the approach seem tailored to the application to a specific region in France. In particular, seasonality is treated by dividing the year into two seasons, one in which extremes typically occur. Plus the model is fitted conditional on one of three possible weather patterns (WPs), based on the spatial correlation of rainfall for the region. Although the number of seasons and WPs could certainly be varied to model rainfall for other types of climate, it is not clear that the constraint of being limited to a quite small number of seasons and WPs could always permit an adequate fit.*

⇒ We think there is a misunderstanding here. We fully agree that the seasonal and weather pattern modeling may not be relevant in other region of the world, but this is precisely what we claim p. 24 lines 29-33: "Although our procedure of selection is general and could be applied to any region of the world -and possibly to other variables (temperature etc)-, we stress that our conclusions are in themselves not universal. In particular, other marginal distributions may be more suitable than the Gamma in other regions of the world showing less or more heavy tails. Although the mixture of distributions over weather patterns has revealed efficient in other countries (e.g. in Norway and Canada, Brigode et al., 2014; Blanchet et al., 2015), it might be less relevant in, e.g., monsoon climate regions where the consideration of seasons seems essential and might be enough." So the term "general" refers to the selection procedure (i.e. to the cross-validation framework), not to the modeling. We claim that the proposed framework, as being based on objective criteria, is general and may be used to select among any distribution. Here we use mixtures of Gamma over seasons and weather patterns, but any other distribution may be considered. The proposed cross-validation criteria are general and independent on the choice of distributions.

In order to clarify this in the article, we propose to add p. 2 line 33 the following sentences: "... in very short distance. Following previous studies in the region (Evin et al., 2016; Garavaglia et al., 2010, 2011; Gottardi et al., 2012), the compared marginal distributions involve seasonal and weather pattern subsampling, considering different models for the subclass-dependent distributions. However the proposed cross-validation

framework is general, as involving objective criteria, and could likewise be used to select among any other distribution.".

*Another potential limitation concerns the performance of the different forms of distributions fitted to rainfall amounts, particularly for extremes. Conclusions are drawn about "heavy tails" that could benefit by relying more on extreme value theory. The restriction to a single distribution may have distorted the performance for extremes, with some of the conclusions conflicting with results in the literature when only extremes are modeled.*

$\Rightarrow$ By "the restriction to a single distribution", we guess you refer to using a single distribution for modeling the whole range of rainfall. As already stated p. 2 lines 18-20, another possibility would indeed be to consider hybrid distributions built by mixing a first distribution representing the bulk of the distribution and a second one focusing on the upper tail and relying on Extreme Value Theory (see the review of Scarrot and MacDonald, 2012, for example). For example Frigessi et al. (2002) consider a mixture of Weibull and Generalized Pareto distributions modeling respectively the bulk and the heavy tail of the distribution. Frigessi's model presents the advantage of removing the delicate choice of a predetermined threshold. However, as noted by Naveau et al. (2016), it also has many parameters (6) which are difficult to estimate. We don't think that applying such a model in the context of WP and season subsampling is relevant. This is why we preferred considering the extended Generalized Pareto model of Naveau et al. (2016) which i) allows to model both the bulk and the tail of the distribution, ii) is in compliance with Extreme Value Theory, iii) is much more parsimonious (3 parameters). Given the lack of robustness of the extended Generalized Pareto distribution (see the SPAN100 scores of Figure 6), we firmly believe that Frigessi's model (or equivalent) would lack robustness even more. Regarding the fact that the selected model is not in compliance with Extreme Value Theory, let us point out that the mixture of Gamma distribution looks actually "pretty much" like the extended Generalized Pareto distribution (compare Figure 7 to Figure 5) but it is much more robust. Finally let us remind that Extreme Value Theory is an asymptotic theory, so it applies to peaks over infinite threshold, which is obviously never the case in practice. However for long-enough data one can usually reasonably assume convergence to the asymptotic case and base analysis of extremes on Extreme Value Theory. So the fact that the distribution founded by Extreme Value Theory is not selected in our case might be an indication that the available data are not long-enough to consider the asymptotic theory to hold.

*For these reasons, I recommend that the manuscript be accepted for publication subject to revision.*

*SPECIFIC COMMENTS*

*(1) Generality of proposed approach*

*It seems like a crude approximation to consider only two seasons and assume stationarity within a given season. More realistic approaches include allowing the parameters of the rainfall distribution to gradually*

*change depending on the time of year. Some regions of the world even have more than one wet season, indicating a limitation of the proposed approach.*

⇒ Actually we assume stationarity within a given season **and** weather type. Figure 4 of the paper shows there is a clear gain in considering the weather patterns (WPs) to complement the information brought by subsampling into seasons. This said, we fully agree that considering two fixed seasons might not be relevant in other region of the world. This is stated in the discussion section, see p. 24 line 31 to p. 25 line 1: "Although the mixture of distributions over weather patterns has revealed efficient in other countries (e.g. in Norway and Canada, Brigode et al., 2014; Blanchet et al., 2015), it might be less relevant in, e.g., monsoon climate regions where the consideration of seasons seems essential and might be enough." However the use of fixed seasons and WPs has already been extensively studied and validated in southern France in other studies (Evin et al., 2016; Garavaglia et al., 2010, 2011; Gottardi et al., 2012). Considering rainfall distributions that gradually change depending on the time of year would be another possibility but it would be tricky in our case. Figure 1 shows the monthly averages of daily nonzero rainfall in the region. It clearly shows the occurrence of a wetter season spanning the months of September, October and November. These three months compose the season-at-risk considered in this paper. A second maximum, although much lower, is found in April and May. Modeling the monthly fluctuations of Figure 1 as a parametric function of time (for example using cos and sin terms) doesn't seem easy to us. Fitting one distribution per month would be another possibility but it would very likely lack robustness by requiring estimating 12 distributions. Considering WPs is an alternative way of modeling the monthly fluctuations through the monthly occurrence of the WPs (see Figure 1). It shows the advantage of being physically-based since the original WPs of Garavaglia et al. (2010) are constructed by clustering synoptic circulations (geopotential fields). Considering WPs allows in particular accounting for the occurrence of two wetter periods -firstly in autumn (season-at-risk) and secondary in spring (in the season-not-at-risk)-, which correspond to larger probabilities of occurrence of WP1, which is the wettest WP for both seasons (see Figure 10 of the article for the case of the season-at-risk). This shows the advantage of being relatively parcimonious (it requires estimating 6 distributions in the WP/season subsampling case).

To make this clearer, we propose to replace Table 4 by a new Figure 4 corresponding to Figure 1 of this response and to replace p. 14 lines 14-16 by: "The occurrence statistics of the three WPs for the period 1948-2013 are presented in Figure 4. The yearly occurrence of the three WPs is roughly similar (27% for WP1, 36% for WP2, 37% for WP3). However the WPs show very different seasonality. In particular WP1 is more frequent in spring and autumn, which correspond to wetter periods, particularly in autumn (see the monthly averages of nonzero rainfall in Figure 4). WP3 is more frequent in summer, which is the driest season, while WP2 features almost a reversed seasonality compared to WP3. This shows that, although being based on the spatial dependence, the WPs are linked to the seasonality of rainfall in the region. We also propose to clarify that the WPs are physically-based by adding p. 14 line 11: "... classification described in

[Figure]

[Figure]

Figure 1: Left: Boxplot of the monthly averages of daily nonzero rainfall. Each boxplot contains 42 points (one point per station). Right: Monthly percentage of occurrence of the three WPs.

Garavaglia et al. (2010), which is obtained by clustering synoptic situations (geopotential heights) for France and surrounding areas into eight classes".

*Conditioning on a few WPs based on the degree of spatial correlation of rainfall is an intriguing and not very common approach. Alternatives in the literature have included either introducing a hidden state variable (likewise assuming only a few possible states), which would require much more involved calculations, or including an observed covariate (such as an index of atmospheric circulation), which would require its identification for a given region but could assume effectively infinitely many possible states. Other than convenience, the advantages of the proposed approach are not clear.*

⇒ This is perfectly right: as stated p. 14 lines 12-13, the three considered WPs are obtained by grouping the eight WPs of Garavaglia et al. (2010) into three classes based on their spatial correlation. However let us recall that the spatial correlation is only a secondary ingredient in the construction of the WPs. The first ingredient is the similarity in synoptic situations since the eight WPs of Garavaglia et al. (2010) are obtained by clustering geopotential fields. Thus the used WPs do already include atmospheric information. This said, we understand it might seem intriguing to consider the spatial correlation for the subgouping whereas this article deals with marginal distribution. Actually this choice is guided by external constraints that we deliberately omitted to mention to not confuse the reader. Indeed, this study is part of a bigger project aiming to build a stochastic daily rainfall generator in the region. The first step of this project was to select an appropriate marginal distribution that could reasonably fit the whole range of rainfall values anywhere within the region. This has led us to develop the cross-validation framework of this article. However having

in mind that the final goal will be to generate rainfall fields and that this generation will be based on WPs, we decided to reduce the number of parameters of the rainfall generator by grouping some of the WPs based on the spatial correlation of rainfall during these WPs. The maps of Figure 10 of the paper lead us conclude that this grouping, although based on the spatial correlation, makes actually good sense for the marginal distribution. There may be other possible groupings but we repeat that the goal of this paper is *not* to find the best marginal model but to propose an objective cross-validation framework for selecting among several marginal distributions.

*(2) Using extreme value theory to interpret results*

*It is concluded that a mixture of gamma distributions provides the best fit, especially for extreme high precipitation amounts. Yet a gamma distribution has a light tail, well known to not be heavy enough for precipitation extremes. Still it is argued that allowing the gamma distribution to vary depending on the season and on the WP induces a heavier tail (Figure 5).*

*If this claim were correct for seasonality, then it would appear that the apparent heavy tail is at least partly an artifact of ignoring seasonality. Yet there is some evidence in the literature (e.g., by explicitly allowing some of the parameters in an extreme value analysis to vary within the year) that this is not necessarily the case.*

*Concerning conditioning on WPs, it is well known that a mixture of gamma distributions can induce a heavier tail than a single gamma. Yet I wonder whether a mixture involving only a few gamma distributions (i.e., only three for the wet season) is sufficient to produce a truly heavy tail (in the sense of extreme value theory).*

*So it may be informative to examine how well the gamma distribution (and the mixture of three gamma distributions depending on the WP) fits precipitation amounts in the wet season alone. As it stands, I worry that the results for extremes may have been distorted by the constraint of fitting a single distribution to all rainfall amounts.*

$\Rightarrow$ This is a very good point. Let consider the marginal distribution of daily rainfall in the season-at-risk. Following Equation (3) of the article, the marginal distribution in season $s$ is given by

$$\mathrm{pr}_s(R \leq r) = p_s^0 + \sum_{k=1}^{K} p_{s,k}(1 - p_{s,k}^0)G_{s,k}(r), \tag{1}$$

where $p_s^0 = \sum_{k=1}^{K} p_{s,k}p_{s,k}^0$ is the probability of any day to be dry in the season $s$. Nonzero precipitation amounts defined by (1) have CDF:

$$G_s(r) = \sum_{k=1}^{K} p'_{s,k}G_{s,k}(r), \tag{2}$$

where $p'_{s,k} = p_{s,k}(1 - p^0_{s,k})/(1 - p^0_s)$. We show in Figures 2 and 3 the cross-validation scores associated to (2) when $s$ is the season-at-risk. Comparing Figure 2 below to Figure 4 of the article reveals the same conclusion, namely also for the season-at-risk there is a strong gain in considering WP subsampling, which applies both for the bulk (NRMSE) and the tail (FF and $N_5$) of the distribution, despite a slight loss in robustness (SPAN$_{100}$). Comparing Figure 3 below to Figure 6 of the article reveals also that, for the season-at-risk as well, the two best distributions are the extended Generalized Exponential (eexp) and the Gamma (gamma) distributions. We propose to add in the article:

- p. 6 line 7 "... $S \times K$ Gamma distributions. Analogously, the CDF of nonzero precipitation amounts in a given season $s$ writes

$$G_s(r) = \mathrm{pr}(R \le r | R > 0, \mathrm{season} = s) = \sum_{k=1}^{K} p''_{s,k} G_{s,k}(r), \tag{5}$$

  where $p''_{s,k} = p_{s,k}(1 - p^0_{s,k})/(1 - \sum_{k=1}^{K} p_{s,k} p^0_{s,k})$."

- p. 20 line 10 "... similarly for Antraigues station. Note that exactly the same conclusions hold when focusing on the season-at-risk rather than considering the whole year, i.e. when computing the cross-validation scores for the estimated seasonal distribution $G_s$ in (5) rather than for the year-round distribution $G$ in (4)."

However, for the sake on concision, we propose not to show any of the Figure 2 or 3 below.

*(3) Assumption of temporal independence*

*It is effectively assumed that the rainfall amounts at an individual site, especially extreme high values, are temporally independent (e.g., second displayed equation on p. 8 and p. 9, line 7). But this assumption never appears to be explicitly stated or verified. There is some evidence in the literature of "clustering" at high levels for time series of daily rainfall amounts at individual sites. Cross validation, depending on how it is implemented, would not properly account for the effects of such temporal dependence.*

$\Rightarrow$ Actually temporal dependence is weak in the region. If we define the wet periods of a given station as the number of consecutive days with nonzero daily rainfall, we find that 43% of the wet periods over the whole region have length 1, 25% have length 2 and 13% have length 3. To focus more on high levels, we computed the extremal index at each station using the method of Ferro and Segers (2003) (R package texmex), considering exceedances above the 90%- and 95%-quantiles. The regional average of the extremal indices amounts 0.65 for the 90%-quantile and 0.68 for the 95%-quantile. This means that the average cluster length of rainfall exceeding high levels is about $1/0.65 \approx 1.5$, which is close to independence (in which case the extremal index is 1). Therefore we think that the hypothesis of temporal independence is defensible for our data. However,

[Figure]

Figure 2: Scores of cross-validation in the season-at-risk when $G_{s,k}$ are Gamma distributions and the number of WT varies: $K \in \{1, 3\}$. The values of $(S, K)$ are indicated in the x-labels. Each boxplot contains 100 points.

[Figure]

Figure 3: Scores of cross-validation in the season-at-risk when $G_{s,k}$ is either the extended exponential (eexp), extended Generalized Pareto (egp), Gamma (gamma), lognormal (lnorm) or Weibull (wei) distribution, with $S = 2$ and $K = 3$. Each boxplot contains 100 points. The boxplots of reliability scores in the lognormal case are missing because they lie far above the upper range of depicted values.

[Figure]

Figure 4: Scores of cross-validation obtained when split sampling blocks of five consecutive days, when $G_{s,k}$ are Gamma distributions and the number of seasons and WT varies: $S \in \{1,2\}$ and $K \in \{1,3\}$. The values of $(S, K)$ are indicated in the x-labels. Each boxplot contains 100 points.

in order not to be biased in the cross-validation procedure by possible weak temporal dependence, we re-made all the estimation but split sampling for the cross-validation blocks of five consecutive days rather than individual days (still imposing half the data to be in the calibration sample and the remaining half to be in the validation sample). Results are shown in Figures 4 to 9, corresponding respectively to Figures 4 to 8 of the article. All results are actually almost similar to those of the article. Therefore the same conclusions hold, namely that i) there is some gain in considering subsampling into seasons and WPs (Figure 4 below), ii) the Gamma and extended Exponential models give overall the best scores of cross-validation, iii) the bivariate thin plate spline with drift in smoothed altitude (tps2Z) is the best interpolation method.

We propose to replace line 1 of p. 15 by: "1. We divide the days of 1948-2013 into two subsamples of equal size, denoted $C^{(1)}$ and $C^{(2)}$. Given the weak temporal dependence of rainfall in the region (80% of the wet periods have length lower than 3), division is made by randomly choosing blocks of five consecutive days to compose $C^{(1)}$, the remaining blocks composing $C^{(2)}$. Figures 4 to 8 of the article will be replaced by the corresponding figures.

[Figure]

Figure 5: Case of Antraigues obtained when split sampling blocks of five consecutive days, when $G_{s,k}$ are Gamma distributions and the number of seasons and WT varies: $S \in \{1, 2\}$ and $K \in \{1, 3\}$. The values of $(S, K)$ are indicated in the title. The dotted lines show the 95%-envelope of return level estimates over the 100 subsamples. The plain line shows the median estimates. The gray points show the full sample (35 years). Each estimation is based on half of these points.

[Figure]

Figure 6: Scores of cross-validation obtained when split sampling block of five consecutive days, when $G_{s,k}$ is either the extended exponential (eexp), extended Generalized Pareto (egp), Gamma (gamma), lognormal (lnorm) or Weibull (wei) distribution, with $S = 2$ and $K = 3$. Each boxplot contains 100 points. The boxplots of reliability scores in the lognormal case are missing because they lie far above the upper range of depicted values.

[Figure]

Figure 7: Case of Antraigues when when split sampling block of five consecutive days, when $G_{s,k}$ is either the extended exponential (eexp), extended Generalized Pareto (egp), lognormal (lnorm) or Weibull (wei) distribution, with $S = 2$ and $K = 3$. The dotted lines show the 95%-envelope of return level estimates over the 100 subsamples. The plain line shows the median estimates. The gray points show the full sample (35 years). Each estimation is based on half these points. Case of the Gamma distribution is shown in the right panel of Figure 5.

*EDITORIAL COMMENTS*

*(1) p. 3, lines 16-17*

*Not clear how the "factor" is defined or calculated.*

$\Rightarrow$ We apologize for the confusing formulation. By "a factor 1 to 2.6 is found for the annual totals of daily rainfall", we mean that the largest average of annual total of daily rainfall is 2.6 times larger than the lowest average of annual total (max= 2111 mm/year, min= 805 mm/year). To make it clearer, we propose to replace p. 3 lines 16-17 by: "... rainfall distribution. To illustrate these disparities, we show in Figure 2 the averages of annual totals and annual maximum daily rainfalls for each station. Computing the ratios between the largest and lowest values in Figure 2 gives a ratio of 2.6 for the annual totals and 3.2 for the annual maxima. For comparison the latter ratio is barely lower than the ratio found over the whole of France, which amounts 4. For both annual totals... "

*(2) p. 25, Figure 10*

*Three of the graphs are for the same quantity, mean of non-zero rainfall for different weather patterns. But the color coding varies making comparisons difficult.*

$\Rightarrow$ We show in Figure 10 the equivalent of Figure 10 of the article when using the same color scale for all the mean maps. It hinders visualizing the regional disparities in WP2 and 3. Therefore we prefer leaving Figure 10 of the article as it is.

[Figure]

Figure 8: Scores of mapping obtained when split sampling block of five consecutive days, when $G_{s,k}$ are Gamma distributions with $S = 2$ and $K = 3$ whose parameters are interpolated with the mapping models of Table 3 of the article. The two first rows show leave-one-out cross-validation scores. Each boxplot contains 200 points. The third row compares interpolations at a given station whether the data of this station are used or not in the interpolation. Each boxplot contains 100 points.

[Figure]

Figure 9: Case of Antraigues (top) and Mayres (bottom) when when split sampling block of five consecutive days, when $G_{s,k}$ are Gamma distributions with $S = 2$ and $K = 3$ whose parameters are interpolated with either: kriging without extrenal drift (krig), stepwise linear model (steplmZ), bivariate thin plate spline with drift (tps2Z), or trivariate thin plate spline (tps3Z). The dotted lines show the 95%-envelope of return level estimates over the 100 subsamples. The plain line shows the median estimates. In black, each interpolation is based on half the data of the other stations, excluding the considered station. In red, interpolation is based on half the data of all the stations, including the considered station. The gray points show the full sample (35 years for both stations).

[Figure]

Figure 10: Map of the probability of daily rainfall to exceed 1mm and of the mean of nonzero rainfall in the three WPs of the season-at-risk. The points are colored with respect to the empirical estimates.

**References**

Blanchet, J., Touati, J., Lawrence, D., Garavaglia, F., and Paquet, E. (2015). Evaluation of a compound distribution based on weather patterns subsampling for extreme rainfall in Norway. *Natural Hazards and Earth System Sciences*, 15(12):2653–2667.

Brigode, P., Bernardara, P., Paquet, E., Gailhard, J., Garavaglia, F., Merz, R., Mićović, Z., Lawrence, D., and Ribstein, P. (2014). Sensitivity analysis of SCHADEX extreme flood estimations to observed hydrometeorological variability. *Water Resources Research*, 50(1):353–370.

Evin, G., Blanchet, J., Paquet, E., Garavaglia, F., and Penot, D. (2016). A regional model for extreme rainfall based on weather patterns subsampling. *Journal of Hydrology*, 541(B):1185–1198.

Ferro, C. A. T. and Segers, J. (2003). Inference for Clusters of Extreme Values. *Journal of the Royal Statistical Society. Series B (Statistical Methodology)*, 65(2):pp. 545–556.

Frigessi, A., Haug, O., and Rue, H. (2002). A dynamic mixture model for unsupervised tail estimation without threshold selection. *Extremes*, 5(3):219–235.

Garavaglia, F., Gailhard, J., Paquet, E., Lang, M., Garçon, R., and Bernardara, P. (2010). Introducing a rainfall compound distribution model based on weather patterns sub-sampling. *Hydrology and Earth System Sciences*, 14(6):951–964.

Garavaglia, F., Lang, M., Paquet, E., Gailhard, J., Garçon, R., and Renard, B. (2011). Reliability and robustness of rainfall compound distribution model based on weather pattern sub-sampling. *Hydrology and Earth System Sciences*, 15(2):519–532.

Gottardi, F., Obled, C., Gailhard, J., and Paquet, E. (2012). Statistical reanalysis of precipitation fields based on ground network data and weather patterns: Application over French mountains. *Journal of Hydrology*, 432–433(0):154 – 167.

Naveau, P., Huser, R., Ribereau, P., and Hannart, A. (2016). Modeling jointly low, moderate and heavy rainfall intensities without a threshold selection. *Water Resources Research*.

Scarrot, C. and MacDonald, A. (2012). A review of extreme value threshold estimation and uncertainty quantification. *REVSTAT - Statistical Journal*, 10(1):33–60.

---

## Author Comment (AC2) · 3 Jul 2018

We warmly thank the Reviewer for his/her valuable comments. We provide below a detailed point-to-point reply to these comments. The proposed changes for the next version of the article are indicated in red.

*The paper describes a method for mapping distribution parameters from single rain gauge record across a domain, but lacks a proper discussion of the role/significance of the proposed framework in the landscape of rainfall hazard mapping: in that respect, if the proposed framework is really a (significant) step forward then the authors should demonstrate it by comparing it to the state-of-the-art in rainfall hazard mapping, which they as such also describe in the introduction (page 2, line 4-14). I am left with the impression of reading a technical report rather than a paper significantly advancing the field (which does not mean there is no advancement per se, but it is difficult to judge at this stage).*

⇒ We think there is a misunderstanding here. First, the idea of mapping rainfall hazard by mapping the parameters of rainfall CDF is not new. See for example Beguería and Vicente-Serrano (2006); Beguería et al. (2009); Szolgay et al. (2009); Blanchet and Lehning (2010); Ceresetti et al. (2012), which are cited p. 2 line 25 of the article. Second, the goal of this paper is not to propose a new way of mapping rainfall hazard, but to propose an objective framework, based on cross-validation criteria, in order to assess goodness-of-fit of the full procedure of hazard mapping. This comprises the selection of both the best marginal distribution and the best mapping method *among those tested*. Thus, we do not claim to propose a new and better way of mapping rainfall hazard, but we propose a framework for selecting objectively among a bench of methods/models. To the best of our knowledge, such a framework has never been proposed so far. As explained p. 2 lines 15-27, usually either a given rainfall CDF is assumed and different mapping methods are compared (often not in a cross-validation framework), or different CDFs are compared but the mapping step is not considered. Here we do both and for this we propose a new cross-validation framework based on objective criteria.

Regarding the recommendation of comparing our results with a method based on interpolating rainfall, as described page 2 line 4-14, this would indeed be interesting but we think this is out of the scope of the paper. As already said, our goal is to develop an objective cross-validation framework for mapping rainfall hazard, not to develop a better way of mapping hazard than state-of-the-art methods. Second, there is not a unique way of spatially interpolating rainfall, and this issue can be the subject of an article on its own (see e.g. Camera et al., 2014; Creutin and Obled, 1982; Goovaerts, 2000; Ly et al., 2011; Rogelis and Werner, 2013, , which are cited p. 2 lines 7-8).

*I do not understand why classic interpolation of rainfall of a certain frequency comes along with issues of zero values (even more obvious when mapping amounts and their exceedance probabilities), see page 2, line 10-11. A certain exceedance probability of rainfall is >0 by definition, and we are talking about regional hazard maps and not about scenarios (i.e. rainstorms). Or am I getting something wrong? Also, provided the*

*issue exists, the authors address a solution themselves (which is the analytical transformation), which brings me back to the issue of ideally comparing the proposed framework to the state-of-the-art.*

⇒ By "spatial interpolation of rainfall", we mean here interpolating every daily raingage value within the region, in order to have daily time series at every grid point of the region. Then a CDF could be estimated at every grid point based on the interpolated time series. Maps such as those of Figures 10 and 11 of the article could directly be produced based on these estimated CDFs. The issue in doing this is that 65% of the daily values are zeros. Interpolating methods work usually well for smoothly varying unbounded values (e.g. temperature), but it performs much more poorly for handling spatial intermittency.

The reviewer mentions the possibility of interpolating probabilities rather than daily rainfall values. This would indeed be an alternative way of doing if a single probability were of interest, for example the 100-year return level. However this is not possible if one aims to be able to produce any probability map. This is for example the case in rainfall simulation frameworks, e.g. when rainfall are input of spatially distributed hydrological models: one needs to be able to simulate any rainfall with the right frequency. In such a case, one needs to know the rainfall CDF at any grid point. To do that, as explained in the article p. 2 lines 3-26, two methods are the great majority of the time followed, either by interpolating rainfall values, or by estimating the CDF at every raingage and then interpolating the parameters of these CDFs. This work follows the latter alternative.

*I also do not understand why one would feed a distributed rainfall-runoff-model with a rainfall-frequency map (page 1, line 17-18). The resulting rainfall-runoff-model output is highly artificial, not much telling about a realistic hydrological scenario. In hydrological hazard/risk assessment, one would probably conduct scenario-based analyses based on potential (realistic) rainstorms or continuously simulate rainfall time series to feed an RR-model, to get insights about (extreme) runoff events. But maybe I did not understand what the authors intend to say here; citations of related work would maybe clarify.*

⇒ Of course we do not intend to feed a distributed rainfall-runoff-model with a rainfall-frequency map. We mean page 1 lines 17-18 that when rainfall-runoff-model are feeded with rainfall simulations, one needs to be able to simulate possible rainfall fields. For this two ingredients are necessary: first to know the rainfall CDF at any grid point, i.e. the marginal distribution. Second to know the spatial correlation of rainfall, i.e. the spatial distribution. This article intends to contribute to the first point, the marginal distribution.

In order to make this clearer, we propose to replace p. 1 lines 18-19 by: spatially distributed hydrological models. In such a case one needs to be able to simulate any possible rainfall fields. This implies knowing both the local occurrence of any rainfall value with the right frequency, and not only the largest ones, and their spatial co-occurrence.

*Modelling only two seasons is a clear limitation, and so is the assumption of stationarity. The same*

*applies to the three weather patterns, which is another constrain. I am also in doubt that the Gamma (or the mixture of a Gamma) is suitable in other regions, especially in the tail. That is, it remains open whether the framework is really applicable to other regions. The authors put that into question themselves (e.g. page 24, line 30, among others). So besides comparing the proposed method to the state-of-theart, a second study area (other climate, more seasons) would be – in my eyes - very important.*

⇒ We would like to stress again that the goal of this article is *not* to propose a new and better way of mapping rainfall hazard. Our goal is to propose an objective framework for selecting among a bench of methods/models in order to pass from isolated raingage records to CDFs maps in a region. Therefore we do not claim that the considered mixture of Gamma distribution is the best model for the region, and even less for other regions of the world. What is claimed is that, based on the proposed cross-validation framework, the mixture of Gamma is the best model *among those tested*. Therefore, although testing the generality of the mixture of Gamma to fit rainfall in other regions of the world would be very interesting, we think this is out of the scope of this paper.

*What is also missing is a proper discussion of the uncertainties of the rainfall records in mountain regions. It would be important to consider these observation errors in the framework, again for the proposed model and the state-of-the-art in comparison, to really understand all implications.*

⇒ This is a very interesting point. Every measurement comes with some uncertainty - whatever the variable of interest. Uncertainty in precipitation measurement is usually larger for snow events, and particularly when wind blows. However the region of interest, although mountainous, is located in a relatively warm area where snowing days are very rare (maximum a few days per year but usually zero). Therefore we believe that data uncertainty in our case can be considered as reasonably small. How small it is and what is its impact on the estimated rainfall distribution? We are sorry to admit we lack information to document this.

*A figure describing the entire framework would be important, it is difficult to follow all steps and practitioners will for sure appreciate a presentation that is a tad more "handson".*

⇒ We thank the reviewer for this nice suggestion. The schematic summary of the procedure is displayed in Figure 1. We propose to start Section 3.3.1 with the sentence: "The full cross-validation procedure for selecting both the marginal and mapping models is summarized in Figure 5.", where the new Figure 5 shows Figure 1 below.

*I would recommend a final proofread by a native speaker, there is quite a number of minor language related mistakes throughout the manuscript, not a big deal but just a few examples: "One of the difficulty"; "models for nonzeros rainfall"; "Similar idea is used"; "in the same time"; "independently on each others"*

⇒ We warmly thank the reviewer for pointing out these typos that will be corrected as follows: one of

[Figure]

Figure 1: Schematic summary of the full cross-validation procedure for selecting both the marginal and mapping models.

the difficulties; models for nonzero rainfall; a similar idea is used; at the same time; independently of each others.

*Sharing the original observation data (other journals even demand it) to allow for reproducing the results is recommended.*

⇒ The rainfall data were provided by Electricité de France and Météo-France. The Météo-France data used in this publication are available through the Public Data portal of Météo-France available at https://publitheque.meteo.fr/okapi/accueil/okapiWebPubli/index.jsp and are free of use for research purpose. The people interested in the Electricité de France data for research purpose should contact Emmanuel Paquet, emmanuel.paquet@edf.fr.

*In summary, at this stage, I am in doubt that the reader really understands the added value of the method, why and when established rainfall hazard mapping methods are competitive and if the method is applicable to another region/climate.*

⇒ We hope we have clarified the added value of this article.

**References**

Beguería, S. and Vicente-Serrano, S. M. (2006). Mapping the hazard of extreme rainfall by peaks over threshold extreme value analysis and spatial regression techniques. *Journal of Applied Meteorology and Climatology*, 45(1):108–124.

Beguería, S., Vicente-Serrano, S. M., López-Moreno, J. I., and García-Ruiz, J. M. (2009). Annual and seasonal mapping of peak intensity, magnitude and duration of extreme precipitation events across a climatic gradient, northeast spain. *International Journal of Climatology*, 29(12):1759–1779.

Blanchet, J. and Lehning, M. (2010). Mapping snow depth return levels: smooth spatial modeling versus station interpolation. *Hydrology and Earth System Sciences*, 14(12):2527–2544.

Camera, C., Bruggeman, A., Hadjinicolaou, P., Pashiardis, S., and Lange, M. A. (2014). Evaluation of interpolation techniques for the creation of gridded daily precipitation (1x1 km2); cyprus, 1980–2010. *Journal of Geophysical Research: Atmospheres*, 119(2):693–712. 2013JD020611.

Ceresetti, D., Ursu, E., Carreau, J., Anquetin, S., Creutin, J. D., Gardes, L., Girard, S., and Molinié, G. (2012). Evaluation of classical spatial-analysis schemes of extreme rainfall. *Nat. Hazards Earth Syst. Sci.*, 12:3229–3240.

Creutin, J. D. and Obled, C. (1982). Objective analyses and mapping techniques for rainfall fields: An objective comparison. *Water Resources Research*, 18(2):413–431.

Goovaerts, P. (2000). Geostatistical approaches for incorporating elevation into the spatial interpolation of rainfall. *Journal of Hydrology*, 228(1):113 – 129.

Ly, S., Charles, C., and Degré, A. (2011). Geostatistical interpolation of daily rainfall at catchment scale: the use of several variogram models in the ourthe and ambleve catchments, belgium. *Hydrology and Earth System Sciences*, 15(7):2259–2274.

Rogelis, M. C. and Werner, M. G. F. (2013). Spatial interpolation for real-time rainfall field estimation in areas with complex topography. *Journal of Hydrometeorology*, 14(1):85–104.

Szolgay, J., Parajka, J., Kohnová, S., and Hlavčová, K. (2009). Comparison of mapping approaches of design annual maximum daily precipitation. *Atmospheric Research*, 92(3):289 – 307. 7th International Workshop on Precipitation in Urban Areas.

---

## Referee Report (RR1)

**An objective cross-validation framework for mapping rainfall hazard based on rain gauge data**

*Juliette Blanchet, Emmanuel Paquet, Pradeebane Vaittinada Ayar, and David Penot*

The manuscript proposes an objective framework for mapping rainfall hazard in an area. It aims at both evaluating the best statistical distribution at station location and assessing the best mapping method. The authors propose the adoption of a unique distribution for the full distribution paying a particular attention to the tails. A small catchment in France is used for assessing the potentialities of the proposed techniques.

The manuscript significantly improved compared to the first version, but I think some of the key-points underlined from both Reviewer#1 and Reviewer#2 are still open, and require to be more carefully assessed for making the manuscript ready for publication.

MAJOR COMMENT

The major issue is related to the aim of the manuscript.

In the answers to the reviewers (R#1 answer 1, R#2 answer 1), and in the manuscript itself (e.g., P1L1), the authors declare that the aim of their work is not to assess the best probability distribution or mapping method in the Ardèche catchment they are analysing, but to propose an effective operational framework for choosing the best approach for mapping rainfall hazard.

Despite this, the results and conclusions sections are not focused on assessing the efficiency of the method in discriminating and choosing the best distribution-mapping method pair for a certain region, but on the description of the results for the case study. As the case study is related to a unique region, and that no-other selection approach or regional study is proposed as comparison and feedback, I can not understand how the author can assess the efficiency of the proposed framework.

I strongly agree with R#2, when he says (answer1) that the authors should do a better work in underlying and demonstrating that the proposed framework is a really significant step forward compared to the state-of-the-art in hazard-mapping. The authors are stating their aim is not to propose a new way of mapping rainfall hazard, but at least, considering that the proposed framework is not particularly innovative from the methodological point of view (as it consists in chaining two standard cross-validation procedures using some regional statistics to verify the combination that provides the best score), I think it's quite important to analyse the common approach adopted for this operation in the literature, and assess the improvement provided by the author's methodology.

Even the effects of one of the stronger assumption adopted from the author in their methodology ( the use of an unique function for the full distribution) are not effectively assessed and just based on some consideration derived from the literature (R#1 answer2).

Concluding, I think the authors should more strongly stress the improvements that their framework and the hypothesis they set, can provide. This could be done by comparing it with the classical "not coupled" methodologies, commonly adopted for selecting separately the best distribution and the best mapping method or trying to verify the effect of different configurations of the framework (e.g., adopting hybrid distributions) on the results on their case study. If it is not feasible, they should at least test the technique on other basins, to provide evidence of the ability of the technique to effectively distinguish the best distribution-mapping technique pairs, according to the different characteristics of the basin.

MINOR ISSUES

I think the manuscript still require a final proofread by a native speaker as a number of language mistakes still arises. E.g.:

- P1L6 – I think "inhomogeneities" is more appropriate than "disparities" when refers to spatial distributions.
- P1L18 - "fields" → "field"
- P2L4 – I don't think "learning" is the correct word.
- P2L11 – "methods are able" → "method is able"
- P3L3 – "lower elevated Rhnoe Valley" → Incorrect, please rephrase
- P5L6 – "they are not be considered" → "they are not considered"
- P6L19 – "distribution is an unsupervised way" → "distribution in an unsupervised way"
- P9L11 – I don't think "#" can be used as variable.
- P14L9 and across the manuscript – "450mm…1mm" separate the measurement units from the number "450 mm… 1 mm"

---

## Author Response (AR2)

We warmly thank the Reviewers for their valuable comments. We provide below a detailed point-to-point reply to these comments. The changes included in the new version of the article are indicated in red.

**1 Answer to Reviewer #1**

*GENERAL COMMENTS*

*The revised version of the manuscript is somewhat improved, as the authors have addressed my comments at least to some extent. My main remaining reservation concerns how the results on the fitting of marginal distributions are interpreted in terms of heavy tails.*

*For this reason, I recommend that the manuscript be accepted for publication subject to minor revision.*

*SPECIFIC COMMENTS*

*(1) Interpretation of results concerning heavy tails*

*The interpretation of the results concerning the shape of the upper tail of the distribution of rainfall amounts still suffers from a lack of reliance on what is known from extreme value theory. In particular, the return level plots for the fitted mixtures of gamma distributions (Fig. 7) all appear approximately linear for high values (i.e., indicative of light-tailed, not heavy-tailed, distributions). Apparently, using a mixture of gamma distributions reduces the bias attributable to fitting the entire span of the values of rainfall rather than only fitting the upper tail (i.e., the mixture of gamma distributions does not necessarily have a heavy, or even a heavier, tail in the sense of extreme value theory).*

*Along the same lines, the corresponding return level plots (Fig. 9) for sub-exponential distributions with heavy tails (in either an asymptotic or sub-asymptotic extreme value theory sense) do not necessarily indicate that the fitted models have too heavy a tail. Rather, their lack of fit in the upper tail is at least partially attributable to being based on fitting the entire range of rainfall values.*

*For this reason, I fear that the statement in the Conclusion and discussion section "distributions showing sub-exponential tails (EGP for example) give usually unrealistic return levels" (p. 28, line 18) will be misunderstood. At best, it is only relevant to the situation in which the distribution (or mixture of distributions) is being fitted to the entire range of rainfall values. As such, it does not necessarily imply anything about how heavy the upper tail of the distribution of rainfall amounts actually is.*

⇒ We apologize for the confusion. We agree that, although the mixture of Gamma distribution allows to get larger return levels than the Gamma distribution alone, it is still unable to produce heavy tails in the sense of extreme value theory. We have made this clearer in the first paragraph p. 20 section 4.1, that has been partly rewritten, stating in particular : "However we note that the return level plots of Figure 7 all

appear approximately linear for high values, meaning that none of the Gamma mixtures is able to produce heavy tails in the sense of extreme value theory. It is possible that return levels at extrapolation far beyond the observed return periods are underestimated." We also recall this in the last paragraph of Section 5 p. 28, that has been partly rewritten, stating in particular: "Possible direction of improvement for the study region regards the choice of the marginal distribution. Although the Gamma mixture was selected according to the cross-validation scores, we noted a possible underestimation of return levels at far extrapolation since the model is unable to produce heavy tails in the sense of extreme value theory."

We also agree that the lack of reliability of the extended Generalized Pareto distribution might be partly attributable to fitting the entire range of values. In particular, we state in the article that the FF and $N_T$ scores tend to be too frequently too small with the extended Generalized Pareto distribution (see Section 4.1), implying a systematic overestimation of the probability of occurrence of large values. This leads to very large return level estimates associated to large return periods. Figure 1 shows that, although both models are theoretically equivalent for extreme values (Naveau et al., 2016), the extended Generalized Pareto gives much larger return level estimates than the Generalized Pareto distribution, due to an overestimation of the shape parameter. The median values of the 500-year return levels exceeds 550 mm/day. This is larger than the largest return level with the Generalized Pareto distribution. We have made this clearer by removing the corresponding sentence and adding in the second paragraph of p. 20 section 4.1: "Note that the lack of reliability of the extended Generalized Pareto in the upper tail is at least partially attributable to being based on fitting the entire range of rainfall values, which leads to a systematic overestimation of the shape parameter $\xi$ in Table 1 compared to when fitting a Generalized Pareto distribution on the upper tail of the data (not shown)." Note also that we considered here the most simple version of the extended Generalized Pareto distribution, while other versions may possibly show better performance (see the second paragraph of p. 28 Section 5: "Other possibility includes considering less parcimonious versions of the extended Generalized Pareto distribution (Naveau et al., 2016) to improve reliability in the upper tail.").

*EDITORIAL COMMENTS*

*(1) Table 1*

*The table indicates that CDFs are listed. Yet for some distributions (i.e., gamma and lognormal) the expressions given are actually for probability density functions, not CDFs.*

$\Rightarrow$ Actually Table 1 indicates: "CDF $G(r)$ or density $g(r)$".

[Figure]

Figure 1: Boxplot of the return levels estimates of the 42 stations with (left) the mixture of Generalized Pareto distribution fitted to the 90%-quantile exceedances in each of the 6 subclasses, (right) the mixture of extended Generalized Pareto distributions. Estimations are made on the full data.

**2  Answer to Reviewer #2**

*The manuscript proposes an objective framework for mapping rainfall hazard in an area. It aims at both evaluating the best statistical distribution at station location and assessing the best mapping method. The authors propose the adoption of a unique distribution for the full distribution paying a particular attention to the tails. A small catchment in France is used for assessing the potentialities of the proposed techniques. The manuscript significantly improved compared to the first version, but I think some of the key-points underlined from both Reviewer#1 and Reviewer#2 are still open, and require to be more carefully assessed for making the manuscript ready for publication.*

*MAJOR COMMENT*

*The major issue is related to the aim of the manuscript. In the answers to the reviewers (R#1 answer 1, R#2 answer 1), and in the manuscript itself (e.g., P1L1), the authors declare that the aim of their work is not to assess the best probability distribution or mapping method in the Ardèche catchment they are analysing, but to propose an effective operational framework for choosing the best approach for mapping rainfall hazard.*

*Despite this, the results and conclusions sections are not focused on assessing the efficiency of the method in discriminating and choosing the best distribution-mapping method pair for a certain region, but on the description of the results for the case study. As the case study is related to a unique region, and that no-other selection approach or regional study is proposed as comparison and feedback, I can not understand how the*

*author can assess the efficiency of the proposed framework.*

⇒ We thank the Reviewer for this comment. However we think there is a misunderstanding here. Sections 3.1.2 and 3.2.2 are entirely devoted to the introduction of scores aiming to objectively compare marginal models and mapping methods (based on their reliability and stability). Sections 4.1 and 4.2 of the results (7 pages in total) use these scores to choose the best distribution-mapping method pair for the region. Only Section 4.3 (one page text + two figures) illustrates the selected model for the region. Of course, we could remove Section 4.3 since the main goal of the article is the cross-validation framework but we find it interesting to briefly illustrate some rainfall features from the selected model, although this is not the main scope of the article.

*I strongly agree with R#2, when he says (answer1) that the authors should do a better work in underlying and demonstrating that the proposed framework is a really significant step forward compared to the state-of-the-art in hazard-mapping. The authors are stating their aim is not to propose a new way of mapping rainfall hazard, but at least, considering that the proposed framework is not particularly innovative from the methodological point of view (as it consists in chaining two standard cross-validation procedures using some regional statistics to verify the combination that provides the best score), I think it's quite important to analyse the common approach adopted for this operation in the literature, and assess the improvement provided by the author's methodology.*

⇒ We agree that the idea of fitting a marginal distribution and interpolating the parameters is not new and we have never claimed so. However we are not aware of any study comparing objectively the full procedure going from point measurements to mapped distributions. This is the gap this article intends to fill. We are sorry to admit that we are not aware of the "common approach adopted for this operation in the literature". To the best of our knowledge and as stated in the introduction, several studies have compared marginal distributions, other have compared mapping models, but there is nothing regarding the full procedure. Furthermore, in the great majority of the studies, the scores of comparison are NRMSE, which is also one of our scores but we complement this with several other scores designed for extreme values or/and assessing robustness and stability.

*Even the effects of one of the stronger assumption adopted from the author in their methodology ( the use of an unique function for the full distribution) are not effectively assessed and just based on some consideration derived from the literature (R#1 answer2).*

⇒ We fully agree that using a unique distribution for the full spectrum of values is questionable, as also stated at the very top of p. 6. However let us first point out that in this article mixtures of (single) distributions (over seasons and WPs) are considered. As shown in the results section, this allows a much greater flexibility and reliability than using a unique distribution without mixing, in particular for the tail.

Another possibility would of course be to use hybrid models based on combining distributions for low and heavy amounts. Very honestly, we chose not to consider hybrid models for the sake of concision since many hybrids models have been proposed in the literature (Vrac and Naveau, 2007; Furrer and Katz, 2008; Li et al., 2012) and there seems to be no consensus on which one is to be preferred. However let us recall that the considered distributions/mapping models only aim at illustrating the proposed framework of model selection. We do not think that multiplying the considered distributions/mapping methods would be much helpful in this case. However we fully agree that if one would like to select the best model for his data, hybrid models would definitively for worth of consideration. We have made this clearer in the conclusion section p. 28: "It could be worth considering hydrid models based on combining distributions for low and heavy amounts (Vrac and Naveau, 2007; Furrer and Katz, 2008; Li et al., 2012), although robustness might be an issue." We have also modified the title and partly rewritten the Abstract and the first paragraph of the conclusion in order to better stress that the scope of the paper is the cross-validation framework.

*Concluding, I think the authors should more strongly stress the improvements that their framework and the hypothesis they set, can provide. This could be done by comparing it with the classical "not coupled" methodologies, commonly adopted for selecting separately the best distribution and the best mapping method or trying to verify the effect of different configurations of the framework (e.g., adopting hybrid distributions) on the results on their case study. If it is not feasible, they should at least test the technique on other basins, to provide evidence of the ability of the technique to effectively distinguish the best distribution-mapping technique pairs, according to the different characteristics of the basin.*

⇒ We thank the reviewer for this comment but we are not sure to understand what is meant by "the classical not coupled methodologies". Here the methodology is also not coupled since we first select the marginal distribution, and then we select the mapping method based on the selected marginal distribution (see Figure 5 of the article). Regarding the idea of testing the framework on other basins, of course it is of interest. Actually, we have also applied the cross-validation framework on a much larger basin, the Durance basin (14,000 km$^2$), in South of France. However we did not present the results here because the Ardèche basin is only chosen for illustration purpose, as a way of showing how the framework works and how it allows to make selection of the best model. To make this clearer, we have slightly modified the abstract, conclusion and the title.

*MINOR ISSUES*

*I think the manuscript still require a final proofread by a native speaker as a number of language mistakes still arises. E.g.:*

*- P1L6 – I think "inhomogeneities" is more appropriate than "disparities" when refers to spatial distributions.*

$\Rightarrow$ Done.

*- P1L18 - "fields" $\rightarrow$ "field"*

$\Rightarrow$ Done.

*- P2L4 – I don't think "learning" is the correct word.*

$\Rightarrow$ Replaced by "estimating".

*- P2L11 – "methods are able" $\rightarrow$ "method is able"*

$\Rightarrow$ Done.

*- P3L3 – "lower elevated Rhnoe Valley" $\rightarrow$ Incorrect, please rephrase*

*Rightarrow* "lower elevated Rhône valley" replaced by "Rhône valley".

*- P5L6 – "they are not be considered" $\rightarrow$ "they are not considered"*

$\Rightarrow$ Done.

*- P6L19 – "distribution is an unsupervised way" $\rightarrow$ "distribution in an unsupervised way"*

$\Rightarrow$ Done.

*- P9L11 – I don't think "#" can be used as variable.*

$\Rightarrow$ We have replaced the notation by $card(B_c)$.

*- P14L9 and across the manuscript – "450mm...1mm" separate the measurement units from the number* "450 mm... 1 mm"

$\Rightarrow$ Done.

**References**

Furrer, E. M. and Katz, R. W. (2008). Improving the simulation of extreme precipitation events by stochastic weather generators. *Water Resources Research*, W12439.

Li, C., Singh, V. P., and Mishra, A. K. (2012). Simulation of the entire range of daily precipitation using a hybrid probability distribution. *Water Resources Research*, 48(3). W03521.

Naveau, P., Huser, R., Ribereau, P., and Hannart, A. (2016). Modeling jointly low, moderate and heavy rainfall intensities without a threshold selection. *Water Resources Research*.

Vrac, M. and Naveau, P. (2007). Stochastic downscaling of precipitation: From dry events to heavy rainfalls. *Water Resources Research*, 43(7). W07402.

---

## Author Response (AR3)

We warmly thank the Reviewers for their valuable comments. We provide below a detailed point-to-point reply to these comments. The changes included in the new version of the article are indicated in red.

**1 Answer to Reviewer #1**

*GENERAL COMMENTS*

*The second revised version of the manuscript has satisfactorily addressed my remaining comments. I have only some minor comments of an editorial nature.*

*MINOR/EDITORIAL COMMENTS*

*(1) p. 8, line 23: "Section 3.1.1" instead of "Section 3.1"?*

→ Done.

*(2) p. 9, line 7: "far tail" instead of "very tail"*

→ Done.

*(3) p. 11, line 7: Notation "tr" could be confusing as it usually denotes the trace of a matrix.*

→ Done: replaced by $"transf"$

*(4) p. 15, legend of Fig. 4: Right and left descriptions should be reversed.*

→ Done.

*(5) p. 19, line 7 ("to illustrate the case of an heavy-tailed distribution"): Judging from the pattern of points on the return level plot (Fig. 7), not very strong evidence of a heavy tail.*

$\rightarrow$ We have modified the sentence: We illustrate the quality of the fit for the station Antraigues, located in the very foothills of the Massif Central slope (see Figure 1), which shows among that largest annual maxima (see Figure 2).

(6) p. 22, line 12: Insert "(i.e., a heavier tail)" after "more convex"

$\rightarrow$ Done.

**2    Answer to Reviewer #3**

*The authors have answered to all the questions of my previous review and have clarified some key-points of the manuscript.*

*Despite this, I still think that there is a significant open issue, that should be amended for the manuscript to be ready for publication. I still think that the advances that the proposed framework could provide to the scientific community does not emerge clearly from the manuscript: the objective framework can be quite useful from an operational point of view, but it ends up being a concatenation of standard approaches that consider a combination of standard indices for taking the final decision.*

*The authors state in answer 2 to my previous report that "they are not aware of any study comparing objectively the full procedure going from point measurements to mapped distribution". It could be true, but considering that in the literature there are plenty of methodologies for comparing different parent distribution and different mapping approaches, even separately (the authors list some of them in the manuscript (P2 LL 25-29), for stressing the relevance of their work the authors should do a better job in underling the advantages of using the kind of "unified objective framework" they propose. Is it faster compared to adopting the assessment of the marginal and of the mapping technique separately? Is it more efficient? More powerful in rank the performance of the different models?*

*This point is not explored at all in the manuscript. Section 4.1 and 4.2 are dedicated, as the authors state in answer 1 to my previous report, to describe the results of the application to the case study, analysing the performance of the tested models with the proposed indices. This can be used for showing the significance of the proposed indices, but can this be considered a validation of the framework on its own? As mentioned in*

*my previous revision, if a fair comparison with analogous techniques is not feasible, at least considering the results on a different basin could provide some sort of validation. Despite the basin is considered only for demonstration purposes, showing that the framework is able to provide the best distribution-mapping tool for any considered region, could be a good test for underling the potentialities of the framework in ranking the different considered models according to the input data.*

$\rightarrow$ As suggested by the Reviewer, we have applied exactly the same framework to a much bigger catchment located in another climatic region, in order to show the generality of the framework. The chosen catchment is the Durance in Cadarache $(14,000 \text{ km}^2)$, located in the southern French Alps (see Figure 1). Note that altitude of the region ranges up to 4000 m.a.s.l whereas the highest peak of the Ardèche catchment are around 1500 m.a.s.l. We used 54 stations with data back to 1975. Figure 2 shows that the largest (resp. smallest) annual totals and annual maxima are about 2/3 the values of the Ardèche catchment (Figure 2 of the article). We use the same three weather types (WTs) as in the article. Based on Figure 3, we also define the season-at-risk as the three months of September, October and November (although we could have extended the season-at-risk to December and possibly January). We apply the cross-validation procedure proposed in the article to select both marginals and mapping models.

Figures 4, 6 and 8 show the same cross-validation criteria as in the article. Figures 5, 7 and 9 illustrate the case of the Coursegoules station which receives the largest annual totals (most south-eastern station in Figure 2). Figure 5 shows the benefit of considering subsampling into seasons and/or WTs. The gain in using both seasons and weather types is clearer for $N_5$ than for $FF$. Note that $FF$, $N_5$ and SPAN are lower than in the case of the Ardèche catchment, revealing a more reliable and robust fit of the tail. We select the model with two seasons and 3 WTs although a more parcimonious model could be to consider seasons only. For the Coursegoules station, the model with no WT (cases (1,1) and (2,1)) tend to underestimate the tail (see Figure 5). As for the Ardèche catchment, we select the Gamma marginal distribution but actually the exponential model is almost as good apart for $N_5$. For Coursegoules station the exponential and Gamma models are almost equivalent (see Figures 5 for (2,3) and 7). Finally, based on Figure 8, we select as in the article the bivariate thin plate spline interpolation using the smoothed altitude as covariate (tps2Z), which improves mainly the spatial robustness (TVD and KLD) compared to the kriging method, as illustrated Figure 9 for Coursegoules. Interestingly, we note that both the same marginal and mapping models are selected for the Ardèche and Durance catchment despite their different size and climatology. However we do not claim that these are the best models for any region. Marginal and model section is obviously region-dependent, so the results obtained here cannot be generalized to any region (even in France). Nevertheless we do claim that the proposed framework is general and can be applied to any region.

Although the results for the Cadarache catchment are interesting in that they prove the generality of

framework, we have decided not to include them in the article because it is already quite long and because demonstrating the framework on the Ardèche catchment seems sufficient to us to illustrate how it works.

[Figure]

Figure 1: The Durance catchement. The red points show the location of the stations. The background shows the altitude in gray scale (1km raster cells). The top left insert shows a map of France with the studied region in red. The black lines are the 400 and 800 m.a.s.l. isolines.

[Figure]

Figure 2: Left: Averages of annual totals (mm). Right: Averages of annual maximum daily rainfalls (mm).

[Figure]

Figure 3: Left: Monthly percentage of occurrence of the three WPs. Right: Boxplot of the monthly averages of daily nonzero rainfall. Each boxplot contains 54 points (one point per station).

[Figure]

Figure 4: Scores of cross-validation when $G_{s,k}$ are Gamma distributions and the number of seasons and WP varies: $S \in \{1, 2\}$ and $K \in \{1, 3\}$. The values of $(S, K)$ are indicated in the x-labels. Each boxplot contains 100 points.

[Figure]

Figure 5: Case of Coursegoules when $G_{s,k}$ are Gamma distributions and the number of seasons and WP varies: $S \in \{1, 2\}$ and $K \in \{1, 3\}$. The values of $(S, K)$ are indicated in the title. The dotted lines show the 95%-envelope of return level estimates over the 100 subsamples. The plain line shows the median estimates. The gray points show the full sample (35 years). Each estimation is based on half of these points.

[Figure]

Figure 6: Scores of cross-validation when $G_{s,k}$ is either the extended exponential (eexp), extended Generalized Pareto (egp), Gamma (gamma), lognormal (lnorm) or Weibull (wei) distribution, with $S = 2$ and $K = 3$. Each boxplot contains 100 points. The boxplots of reliability scores in the lognormal case are missing because they lie far above the upper range of depicted values.

[Figure]

Figure 7: Case of Coursegoules when $G_{s,k}$ is either the extended exponential (eexp), extended Generalized Pareto (egp), lognormal (lnorm) or Weibull (wei) distribution, with $S = 2$ and $K = 3$. The dotted lines show the 95%-envelope of return level estimates over the 100 subsamples. The plain line shows the median estimates. The gray points show the full sample (35 years). Each estimation is based on half these points. Case of the Gamma distribution is shown in the right panel of Figure 5.

[Figure]

Figure 8: Scores of mapping when $G_{s,k}$ are Gamma distributions with $S = 2$ and $K = 3$ whose parameters are interpolated with the same mapping models as in the article. The two first rows show leave-one-out cross-validation scores. Each boxplot contains 200 points. The third row compares interpolations at a given station whether the data of this station are used or not in the interpolation. Each boxplot contains 100 points.

[Figure]

Figure 9: Case of Coursegoules when $G_{s,k}$ are Gamma distributions with $S = 2$ and $K = 3$ whose parameters are interpolated with either: kriging without extrenal drift (krig), stepwise linear model (steplmZ), bivariate thin plate spline with drift (tps2Z), or trivariate thin plate spline (tps3Z). The dotted lines show the 95%-envelope of return level estimates over the 100 subsamples. The plain line shows the median estimates. In black, each interpolation is based on half the data of the other stations, excluding the considered station. In red, interpolation is based on half the data of all the stations, including the considered station. The gray points show the full sample (40 years).